# FluxNet: Learning Capacity-Constrained Local Transport Operators for Conservative and Bounded PDE Surrogates

**Zishuo Lan** [1]  **Junjie Li** [1]  **Lei Wang** [1]  **Jincheng Wang** [1]

## Abstract

Autoregressive learning of time-stepping operators provides an effective approach to data-driven partial differential equation (PDE) simulation, yet for conservation laws, they face a fundamental challenge: learned updates may violate global conservation over long rollouts. For the important subclass of mass-conservation-type equations, the problem is compounded by inherent physical bounds (e.g., nonnegativity or concentrations in [0,1]) whose violation further destabilizes predictions. We introduce FluxNet, which learns cumulative transport amounts representing the total conserved quantity redistributed between each cell and a configurable neighborhood over the full surrogate interval. A conservative update guarantees exact discrete conservation by construction; modular capacity-constrained transport heads (L, U, and D) enforce lower bounds, upper bounds, or near-zero dual-bound violations through architectural design. Unlike flux-rate surrogates that require temporal integration and thus inherit CFL constraints, FluxNet involves no such integration; configurable transport neighborhoods enable large-timestep prediction at full spatial resolution. Ghost cells extend the framework to nonperiodic boundaries. Experiments on four benchmarks (1D convection–diffusion, 2D shallow water, 1D traffic flow, 2D Cahn–Hilliard) demonstrate exact conservation, structural bound preservation, architecture modularity, and superior stability over flux-rate surrogates at large temporal strides. The code is publicly available at: https://github.com/Lan-zs/FluxNet.

## 1. Introduction

Autoregressive neural surrogates for partial differential equations (PDEs) offer a promising route to accelerating scientific simulations by replacing expensive numerical solvers with fast neural network inference (Li et al., 2021; Kochkov et al., 2021). In this paradigm a learned time-stepping operator is applied repeatedly to propagate predictions forward in time. A well-known difficulty is that prediction errors accumulate during long-horizon rollouts, gradually degrading forecast quality (Brandstetter et al., 2022; Lippe et al., 2023). For the broad class of PDEs governed by conservation laws—describing phenomena from fluid dynamics and traffic flow (Lighthill & Whitham, 1955) to materials microstructure evolution (Cahn & Hilliard, 1958)—this challenge is compounded by a more fundamental requirement: certain global integrals (mass, momentum, energy, etc.) must remain exactly constant over time in a closed system (Li et al., 2022). For the important subclass of mass-conservation-type equations governing density, concentration, or water depth, the conserved quantity additionally carries inherent physical bounds: mass and water depth must be nonnegative, while concentrations and densities cannot exceed saturation limits (Zhang & Shu, 2010). Standard neural network architectures provide no structural mechanism to enforce either property. As a result, autoregressive rollouts commonly exhibit *conservation drift*, where the total conserved quantity gradually deviates from its initial value, and *bound violations*, where predictions fall outside the physically admissible range. Such errors are not merely inaccurate but unphysical, and can rapidly destabilize long-horizon predictions.

Existing approaches address these issues through soft constraints, post-hoc corrections, or structural flux-based architectures. Soft constraint methods add penalty terms to the training loss (Raissi et al., 2019) but cannot guarantee satisfaction at test time; residual violations accumulate during rollout, in a manner analogous to exposure bias in sequence modeling (Bengio et al., 2015; Ross et al., 2011). Post-hoc projection can enforce conservation or bounds after each step (Singha, 2025; Cardoso-Bihlo & Bihlo, 2025; Liu et al., 2025), but corrections alter local transport structure. A more principled approach is provided by finite-volume-method-

[1] State Key Laboratory of Solidification Processing, Northwestern Polytechnical University, Xi'an, 710072, China. Correspondence to: Junjie Li <lijunjie@nwpu.edu.cn>, Jincheng Wang <jchwang@nwpu.edu.cn>.

*Proceedings of the 43rd International Conference on Machine Learning*, Seoul, South Korea. PMLR 306, 2026. Copyright 2026 by the author(s).

inspired (FVM-inspired) neural surrogates (Praditia et al., 2021; Karlbauer et al., 2022; Horie & Mitsume, 2024; Kim & Kang, 2025; Liu et al., 2026), which predict instantaneous flux rates at cell interfaces between nearest neighbors and update states via divergence-form integration, preserving conservation by construction. However, these flux-rate methods do not address the preservation of physical bounds beyond conservation. Furthermore, they inherit the Courant–Friedrichs–Lewy (CFL) stability condition from the underlying explicit integration scheme, coupling the temporal stride to spatial resolution and limiting achievable speedups. Grid coarsening can relax this coupling but degrades spatial accuracy, an unacceptable trade-off for applications requiring high resolution such as microstructure simulation in materials science.

We propose FluxNet, a neural surrogate that enforces both conservation and physical bounds by construction through the principle of *feasible transport*. FluxNet predicts *cumulative transport amounts*—the total conserved quantity redistributed between each pair of neighboring cells over the entire surrogate interval $\Delta t_{\text{model}}$. Conservation holds because every transport that leaves one cell enters another, so all contributions cancel exactly when summed over the domain. Boundedness is enforced by requiring each transport to be *feasible*: outflow from a cell cannot exceed what it holds above its lower bound, and inflow cannot exceed the receiver's remaining capacity below its upper bound. Because FluxNet directly predicts cumulative transport amounts over the entire surrogate interval rather than integrating instantaneous flux rates through an explicit time-stepping scheme, it does not inherit the CFL stability condition. Configurable transport neighborhoods extending beyond nearest neighbors further decouple temporal and spatial step sizes, allowing large time steps at full spatial resolution. It is worth clarifying that we adopt the name FluxNet to situate our method within the flux-based surrogate literature while emphasizing that the learned quantities are time-integrated cumulative fluxes rather than instantaneous flux rates.

We instantiate this framework through modular, capacity-constrained transport heads. An L-head guarantees strict lower bounds by parameterizing each outgoing transport as a learned fraction of available surplus; a U-head guarantees strict upper bounds by parameterizing each incoming transport as a fraction of remaining capacity; and a D-head addresses dual bounds via two parallel branches coupled with a consistency regularizer, achieving near-zero violations empirically. A ghost-cell treatment extends the framework to non-periodic boundary conditions. We validate FluxNet on four benchmarks: 1D convection–diffusion ($c \geq 0$), 2D shallow water equations (coupled mass–momentum, $h \geq 0$), 1D traffic flow ($\rho \in [0, 1]$ with shocks and rarefaction waves), and 2D spinodal decomposition governed by the Cahn–Hilliard equation ($\phi \in [0, 1]$). Across all tasks,

FluxNet achieves machine-precision conservation, markedly reduced bound violations, and improved long-rollout accuracy. Comparison with FluxGNN (Horie & Mitsume, 2024), a flux-rate-based conservative surrogate, confirms both the modularity of our transport heads (FluxGNN-D improves upon FluxGNN, and FNO backbones benefit from the same heads) and FluxNet's unique stability at large time steps where flux-rate methods diverge.

## 2. Related Work

### 2.1. Neural Operators and Long-Horizon Rollout Stability

Neural operators provide a data-driven approach to learning solution operators of partial differential equations. The Fourier Neural Operator parameterizes integral kernels in Fourier space, enabling resolution-invariant learning for a broad class of PDEs (Li et al., 2021). Physics-informed DeepONets extend operator learning with PDE residual constraints (Wang et al., 2021). Message-passing neural PDE solvers leverage graph neural networks (Gilmer et al., 2017; Battaglia et al., 2018) to handle unstructured meshes (Brandstetter et al., 2022; Pfaff et al., 2020), while multiscale approaches address varying spatiotemporal resolutions (Gupta & Brandstetter, 2023). Machine learning has also been used to accelerate traditional CFD solvers (Kochkov et al., 2021). However, these architectures do not inherently enforce physical conservation laws, and the resulting violation of conservation structure causes predictions to diverge rapidly during autoregressive rollouts (Li et al., 2022). Strategies such as pushforward training (Brandstetter et al., 2022; Sanchez-Gonzalez et al., 2020) and scheduled sampling (Bengio et al., 2015; Ross et al., 2011) mitigate this instability by exposing models to their own predictions during training, yet they provide no formal guarantees on conservation or solution boundedness.

### 2.2. Flux-Based Conservative Neural Surrogates

Flux-based architectures provide a principled approach to structurally preserving conservation in neural surrogates. FINN (Praditia et al., 2021; Karlbauer et al., 2022) adopts an FVM-inspired architecture that decomposes PDE dynamics into constituent physical processes (diffusion, advection, reaction), with learned flux functions at cell interfaces serving as core components for modeling conservation law dynamics. FluxGNN (Horie & Mitsume, 2024) predicts inter-cell flux rates via graph neural networks and achieves exact discrete conservation on unstructured meshes through the divergence-form update structure. Other recent works approximate numerical fluxes for hyperbolic conservation laws using Fourier Neural Operators (Kim & Kang, 2025) or enforce entropy stability in neural flux-form models (Liu et al., 2026).

A common feature of these FVM-inspired methods is that they learn instantaneous flux rates at interfaces between nearest-neighbor cells and compose them via explicit temporal integration to update the state. This formulation inherits the CFL stability condition from the underlying integration scheme, coupling the surrogate's temporal stride to spatial resolution. FluxNet changes the learned quantity from instantaneous rates to cumulative transport amounts exchanged with cells within a configurable neighborhood over the full surrogate interval $\Delta t_{\text{model}}$, eliminating temporal integration and the associated CFL constraint.

### 2.3. Positivity and Bound Preservation

Positivity preservation and boundedness are classical concerns in computational physics, where violations can cause numerical instability or unphysical solutions. Traditional numerical schemes achieve these properties through careful limiter design or convex combinations (Van Leer, 1979; Zhang & Shu, 2010), with finite volume methods providing a natural framework. In machine learning approaches, boundedness is typically enforced via penalty terms, output clipping, or sigmoid squashing, but such strategies interact poorly with autoregressive rollout: small violations accumulate over time, eventually producing catastrophic errors (Liu et al., 2026). The flux-based conservative surrogates reviewed above, while providing exact conservation, do not address the preservation of physical bounds on the conserved quantities. Our capacity-constrained transport heads fill this gap by providing structural bound guarantees: every predicted transport plan yields bounded updates regardless of the learned parameters.

## 3. Method

### 3.1. Problem Setup

We consider conservative PDE dynamics discretized on regular grids. Let the state at discrete time $t$ be $\mathbf{u}^t \in \mathbb{R}^{C \times H \times W}$ (or $C \times L$ in 1D), where each channel represents a conserved quantity such as concentration, density, or momentum. Given optional external fields $\boldsymbol{\xi}$ (e.g., velocity fields), our goal is to learn an autoregressive surrogate operator $\mathcal{T}_\theta$ that maps the current state to the next:

$$\mathbf{u}^{t+1} = \mathcal{T}_\theta(\mathbf{u}^t, \boldsymbol{\xi}), \tag{1}$$

which is rolled out for long-horizon prediction. Standard surrogates that directly predict $\mathbf{u}^{t+1}$ lack structural mechanisms for maintaining conservation or physical bounds, leading to conservation drift and bound violations that compound during rollout. We address this by learning capacity-constrained cumulative transport amounts, which guarantee conservation and bound preservation by construction while eliminating CFL stability constraints.

### 3.2. Conservative Transport Update

Let $\mathcal{N}(i)$ denote the neighborhood of grid cell $i$ defined by a fixed stencil on the grid. We parameterize a directed transport amount $F_{i \to j}$ representing the quantity transferred from cell $i$ to cell $j$ in one surrogate step. The state update follows:

$$u_i^{t+1} = u_i^t - \sum_{j \in \mathcal{N}(i)} F_{i \to j} + \sum_{j \in \mathcal{N}(i)} F_{j \to i}. \tag{2}$$

On regular grids, we organize transport amounts via a stencil channelization inspired by the Lattice Boltzmann Method (LBM): each directional offset $\mathbf{d} \in \mathcal{D}$ in the stencil carries a dedicated transport field representing the cumulative amount $F_{i \to (i+\mathbf{d})}$ transferred from every cell $i$ to its neighbor at offset $\mathbf{d}$, so that a $3 \times 3$ neighborhood in 2D yields 8 directional fields (analogous to the D2Q9 velocity set without the rest channel). Inflow from neighbors is computed by spatially shifting the corresponding outgoing transport fields. We stress that the analogy is purely structural: FluxNet imposes neither collision operators nor equilibrium distributions from LBM (Timm et al., 2016).

**Proposition 1 (Discrete Conservation).** Under periodic boundary conditions, if every directed neighbor relation is symmetric (i.e., $j \in \mathcal{N}(i)$ implies $i \in \mathcal{N}(j)$), then the update above exactly preserves the global sum for each conserved channel: $\sum_i u_i^{t+1} = \sum_i u_i^t$.

The proof follows from summing the update over all cells: the outflow sum $\sum_i \sum_j F_{i \to j}$ and the inflow sum $\sum_i \sum_j F_{j \to i}$ enumerate identical terms up to index renaming and hence cancel exactly. The full proof is given in the appendix. This conservation property holds independently of how the transport amounts $F$ are produced, as long as the same quantity serves as outflow for the sender and inflow for the receiver.

**Remark 1 (Cumulative Transport and CFL-Free Property).** The transport amounts $F_{i \to j}$ in Eq. (2) represent the total conserved quantity redistributed from cell $i$ to cell $j$ over the entire surrogate interval $\Delta t_{\text{model}}$. This differs from FVM-inspired surrogates that learn instantaneous flux rates $f_{i \to j}$ and integrate via $u_i^{t+1} = u_i^t - \Delta t \sum_j f_{i \to j} + \cdots$, where the CFL condition $\Delta t \leq \Delta x / v_{\max}$ limits stability. FluxNet's update (Eq. 2) contains no $\Delta t$ multiplier and performs no temporal integration; the CFL condition therefore does not arise. To accommodate transport spanning $R$ cells during $\Delta t_{\text{model}}$, the stencil radius is set to $R$, decoupling the temporal stride from spatial resolution. All structural guarantees (Propositions 1–3) hold for any neighborhood size and any $\Delta t_{\text{model}}$.

### 3.3. Capacity-Constrained Transport Heads

We produce the directional transport fields via a modular backbone-head factorization. A shared convolutional ResNet-style encoder (He et al., 2016) maps the concatenated input $[\mathbf{u}^t, \boldsymbol{\xi}]$ to feature maps $\mathbf{z} = f_\theta([\mathbf{u}^t, \boldsymbol{\xi}])$. A transport head $h_\psi$ then applies $1 \times 1$ convolutions to $\mathbf{z}$, producing raw parameter fields whose channel count depends on the head type (Table 1). These raw parameters are then transformed via elementwise sigmoid, softmax, or softplus into physically constrained transport quantities. This modular design enables fair comparison: direct regression baselines share the same backbone, and different heads can be swapped without modifying the encoder. Table 1 summarizes the family.

For a stencil $\mathcal{D}$ with $K = |\mathcal{D}|$ directions, the N-head produces $K$ raw parameter fields $\hat{F}_{i \to (i+\mathbf{d})}$, one per direction $\mathbf{d} \in \mathcal{D}$, and uses them directly as signed transport amounts $F_{i \to (i+\mathbf{d})} = \hat{F}_{i \to (i+\mathbf{d})}$. The P-head applies a softplus activation $F_{i \to (i+\mathbf{d})} = \mathrm{softplus}(\hat{F}_{i \to (i+\mathbf{d})})$ to ensure nonnegative transport. Both heads preserve conservation through the update structure (Proposition 1) and serve as general-purpose heads for conserved quantities without physical bound constraints, such as momentum components that can take either sign.

For fields satisfying a lower bound $u_i \geq \ell$, the L-head parameterizes capacity-limited outflow. Define the available amount $a_i = u_i - \ell$ that cell $i$ can transfer without violating the lower bound. The head produces $K+1$ raw output channels, split into a scalar field $\hat{\alpha}_i$ and $K$ directional fields $\{\hat{\pi}_{i,\mathbf{d}}\}_{\mathbf{d} \in \mathcal{D}}$. An outflow fraction $\alpha_i = \sigma(\hat{\alpha}_i) \in (0,1)$ is computed via sigmoid, representing the proportion of available amount to transfer, and distribution weights $\pi_{i \to (i+\mathbf{d})} = \mathrm{softmax}_{\mathbf{d}}(\hat{\pi}_{i,\mathbf{d}}) \geq 0$ with $\sum_{\mathbf{d} \in \mathcal{D}} \pi_{i \to (i+\mathbf{d})} = 1$ are computed via softmax, allocating outflow among neighbors. The outgoing transport amount is $F_{i \to (i+\mathbf{d})} = a_i \cdot \alpha_i \cdot \pi_{i \to (i+\mathbf{d})}$.

**Proposition 2 (L-Head Lower Bound Guarantee).** If $u_i^t \geq \ell$ for all $i$, then after one conservative transport update using the L-head, $u_i^{t+1} > \ell$ for all $i$.

The total outflow from cell $i$ is $a_i \alpha_i < a_i = u_i - \ell$, so $u_i - \mathrm{outflow} > \ell$; since all inflow terms are nonnegative, the bound is preserved. This provides a strict structural guarantee without requiring post-hoc clipping. The detailed proof is given in the appendix.

The U-head enforces an upper bound $u_i \leq u_{\max}$ through a dual construction that limits inflow rather than outflow. It uses remaining capacity $b_i = u_{\max} - u_i$ in place of available amount. The head produces $K+1$ channels with the same sigmoid-softmax scheme: an inflow fraction $\beta_i = \sigma(\hat{\beta}_i) \in (0,1)$ controls what proportion of $b_i$ cell $i$ is allowed to absorb, and softmax weights $\rho_{i,\mathbf{d}}$ distribute

this budget across neighbor directions, giving incoming transport $F_{(i+\mathbf{d}) \to i} = b_i \cdot \beta_i \cdot \rho_{i,\mathbf{d}}$.

**Proposition 3 (U-Head Upper Bound Guarantee).** If $u_i^t \leq u_{\max}$ for all $i$, then after one conservative transport update using the U-head under periodic boundary conditions, $u_i^{t+1} < u_{\max}$ for all $i$.

The proof is dual to that of Proposition 2: total inflow to cell $i$ is $b_i \beta_i < b_i = u_{\max} - u_i$, and all outflow terms are nonnegative. The U-head thus provides a strict structural guarantee for upper bounds analogous to the L-head guarantee for lower bounds.

### 3.4. Dual-Branch D-Head for Double-Bounded Transport

For doubly-bounded fields $u \in [\ell, u_{\max}]$, a single one-sided capacity constraint is insufficient. The D-head produces $2(K+1)$ raw output channels, split into an outflow branch and an inflow branch, each with $K+1$ channels. The outflow branch uses available amount $a_i = u_i - \ell$ with L-head structure, guaranteeing the sender cannot drop below $\ell$. The inflow branch uses remaining capacity $b_i = u_{\max} - u_i$ with U-head structure, guaranteeing the receiver cannot exceed $u_{\max}$. These two branches predict the same physical transport from opposite perspectives.

Each branch computes its own state change: $\Delta u_i^{\mathrm{out}} = -\sum_j F_{i \to j}^{\mathrm{out}} + \sum_j F_{j \to i}^{\mathrm{out}}$ and $\Delta u_i^{\mathrm{in}} = -\sum_j F_{i \to j}^{\mathrm{in}} + \sum_j F_{j \to i}^{\mathrm{in}}$. The final state update averages the two:

$$u_i^{t+1} = u_i^t + \frac{1}{2}\left(\Delta u_i^{\mathrm{out}} + \Delta u_i^{\mathrm{in}}\right). \tag{3}$$

Agreement is enforced via the Dual Consistency Loss (DCL):

$$\mathcal{L}_{\mathrm{DCL}} = \frac{1}{|\Omega|} \sum_i \left|\Delta u_i^{\mathrm{out}} - \Delta u_i^{\mathrm{in}}\right|^2. \tag{4}$$

Each branch individually is conservative and satisfies a single-sided hard bound. The averaged update always guarantees conservation, but simultaneous satisfaction of both bounds holds only when the two branches agree exactly; in practice, DCL training drives violations to near-zero levels with negligible magnitudes. We report violation rates and conditional magnitudes as first-class metrics in all dual-bounded experiments.

### 3.5. Task-Specific LAP Head for Shallow Water Equations

The shallow water equations couple three conserved fields $(h, m_x, m_y)$ where momentum transport depends strongly on mass transport. We introduce FluxNet-LAP, a task-specific instantiation that improves modeling fidelity without changing the core conservative update. Let $h$ be

*Table 1.* Transport heads as feasible-set parameterizations. $K = |\mathcal{D}|$ is the number of neighbor directions.

| Head | Channels | Learned Object | Hard Constraint | Parameterization |
|------|----------|----------------|-----------------|------------------|
| N | $K$ | Signed transport $F_{i \to j}$ | Conservation | Raw conv outputs |
| P | $K$ | Positive outflow | Conservation | softplus$(\cdot)$ on transport |
| L | $K+1$ | Capacity-limited outflow | Conserv. $+ u \geq \ell$ | $\sigma(\cdot)$ outflow %, softmax dist. |
| U | $K+1$ | Capacity-limited inflow | Conserv. $+ u \leq u_{\max}$ | $\sigma(\cdot)$ inflow %, softmax dist. |
| D | $2(K+1)$ | Dual capacity (out/in) | Conservation; dual-bound empirical via DCL | Two branches + DCL |

water depth satisfying $h \geq 0$ and $\mathbf{m} = (m_x, m_y)$ be momentum fields. FluxNet-LAP predicts depth transport using an L-head, producing depth transport amounts $F_{i \to j}^h \geq 0$ that are nonnegative by construction and guarantee $h^{t+1} \geq 0$ (Proposition 2). Momentum transport is decomposed into an advection term carried by mass transport, $F_{i \to j}^{m,\text{adv}} = F_{i \to j}^h \cdot m_i / (h_i + \varepsilon)$, where $\varepsilon > 0$ prevents division issues in near-dry regions, and a pressure term parameterized by a P-head with depth-squared gating, $F_{i \to j}^{m,\text{prs}} = h_i^2 \cdot \text{softplus}(\hat{G}_{i \to j})$, where $\hat{G}_{i \to j}$ denotes the raw output of the pressure branch for direction $(i,j)$. The total momentum transport is $F_{i \to j}^m = F_{i \to j}^{m,\text{adv}} + F_{i \to j}^{m,\text{prs}}$. The $h^2$ gating suppresses spurious momentum exchange in low-depth cells while permitting pressure-driven transport in wet regions. We use $\varepsilon = 10^{-6}$ throughout; sensitivity analysis (Table 8 in the appendix) confirms insensitivity for $\varepsilon \leq 10^{-4}$. LAP is a shallow-water specialization; all other benchmarks use generic heads.

### 3.6. Non-Periodic Boundary Conditions via Ghost Cells

The conservation guarantee of Proposition 1 relies on a symmetric neighborhood under periodic boundaries. For non-periodic domains with prescribed boundary conditions, we introduce a ghost cell treatment, a standard technique in computational physics, that extends FluxNet without modifying the core transport mechanism.

**Ghost cell method.** Given a physical domain $\Omega$ with prescribed boundary values, we pad $R$ ghost cells on each side ($R$ = stencil radius), filled according to the boundary condition type (Dirichlet values, Neumann extrapolation, or Robin combination). A binary identity channel (1 for interior, 0 for ghost) is appended to the input, enabling the network to distinguish boundary from interior cells. The backbone uses replicate padding in place of circular padding. The transport update operates on the extended domain $\hat{\Omega} = \Omega \cup \mathcal{G}$ identically to the periodic case; only interior cell values are retained as the next state (schematic in Figure 9 of the appendix).

**Conservation as flux balance.** On the extended domain $\hat{\Omega}$, Proposition 1 guarantees exact total mass conservation. Since fluxes between interior cells cancel upon summation (by the same index-renaming argument), the interior mass

satisfies

$$\sum_{i \in \Omega} u_i^{t+1} = \sum_{i \in \Omega} u_i^t - F_{\partial \Omega}, \tag{5}$$

where $F_{\partial \Omega}$ is the net flux from interior to ghost (boundary) cells. Interior mass changes only through boundary fluxes, which is the physically correct flux-balance identity for open domains.

### 3.7. Training

All models are trained by minimizing prediction loss between surrogate rollouts and ground-truth trajectories. We employ the pushforward trick (Brandstetter et al., 2022): the model is unrolled for $P$ steps, feeding back its own detached predictions $\hat{\mathbf{u}}^{p+1} = \mathcal{T}_\theta(\text{sg}(\hat{\mathbf{u}}^p), \boldsymbol{\xi})$ with $\hat{\mathbf{u}}^0 = \mathbf{u}^0$, where $\text{sg}(\cdot)$ denotes stop-gradient so that earlier predictions serve as adversarial perturbations of the true input distribution. The training loss is:

$$\mathcal{L} = \frac{1}{|\Omega|} \sum_i \left| \hat{u}_i^1 - u_i^1 \right|^2 + \frac{1}{|\Omega|} \sum_i \left| \hat{u}_i^P - u_i^P \right|^2 + \gamma \mathcal{L}_{\text{DCL}}, \tag{6}$$

where the first term is the one-step loss, the second is the pushforward stability loss, and the DCL term applies when using the D-head. This training strategy mitigates error accumulation, while FluxNet prevents physically inconsistent predictions from compounding during rollout.

## 4. Experiments

We evaluate FluxNet on four representative conservative PDE benchmarks: 1D convection–diffusion (nonnegative scalar), 2D shallow water equations (coupled depth–momentum with dry regions), 1D traffic flow (dual-bounded density with shocks), and 2D spinodal decomposition (dual-bounded concentration field). All benchmarks are evaluated under periodic boundary conditions. The traffic flow benchmark is additionally evaluated under Dirichlet conditions (Section 4.3), serving as the testbed for validating the ghost cell extension (Section 3.6). We compare against ResNet-AR (direct autoregressive regression with identical backbone), FNO-AR (Fourier Neural Operator with residual prediction), soft conservation penalty (+SoftCons), post-hoc projection (+Box+Mass Projection), and output squashing (+SigmoidBound). For direct comparison with flux-based

*Table 2.* Convection–diffusion rollout results at $T = 1$. Conservation error is near machine precision for all flux-based heads.

| Method | $\mathcal{E}_{\text{MAE}}$ ($\times 10^{-3}$) | $\mathcal{E}_{\text{cons}}$ |
|---|---|---|
| FluxNet-N | $16.73 \pm 24.22$ | $3.22 \times 10^{-8}$ |
| FluxNet-P | $1.51 \pm 0.45$ | $4.34 \times 10^{-8}$ |
| **FluxNet-L** | $\mathbf{1.36 \pm 0.41}$ | $2.46 \times 10^{-8}$ |

conservative surrogates, we additionally benchmark against FluxGNN (Horie & Mitsume, 2024), which predicts intercell flux rates via graph neural networks with conservation guarantees. Since FluxGNN is designed for non-periodic boundaries, this comparison is conducted on the Dirichlet traffic flow benchmark. To probe the modularity of our transport heads, we build FluxGNN-D by equipping FluxGNN's backbone with our D-head, and FluxNet-D (FNO) by equipping an FNO backbone with our D-head. Except for the illustrative spinodal decomposition case, which reports results from a single training run, all results are reported as mean $\pm$ standard deviation over 5 independently trained models with different random seeds. We report rollout MAE at $T = 2\times$ training horizon for temporal extrapolation (except convection–diffusion), maximum conservation drift over the rollout, bound violation rate as percentage of spatiotemporal grid points violating bounds, and conditional violation magnitude as mean violation amplitude over violating points only. Additional experimental details are provided in the appendix.

### 4.1. Convection–Diffusion

The 1D convection–diffusion equation with nonnegative concentration ($c \geq 0$) serves as a minimal benchmark for comparing the three transport head variants. Table 2 reports rollout accuracy at $T = 1$. All flux-based heads maintain conservation at machine precision ($\sim 10^{-8}$). The L-head achieves the lowest MAE ($1.36 \times 10^{-3}$), closely followed by the P-head ($1.51 \times 10^{-3}$), while the N-head exhibits highly unstable behavior with an order-of-magnitude higher mean MAE ($16.73 \times 10^{-3}$) and extreme variance across seeds. This instability arises because the N-head permits arbitrary signed transport, allowing simultaneous opposing flows whose cancellation artifacts accumulate during rollout, occasionally producing severe oscillations (see Figure 5 in the appendix). The P-head eliminates this failure mode by constraining transport to be nonnegative via softplus, while the L-head further restricts total outflow to the available amount above zero. Based on the N-head's unreliable performance, it is not used in subsequent experiments.

### 4.2. Shallow Water Equations

The 2D shallow water equations couple three conserved fields: water depth $h$ and momentum components $(m_x, m_y)$.

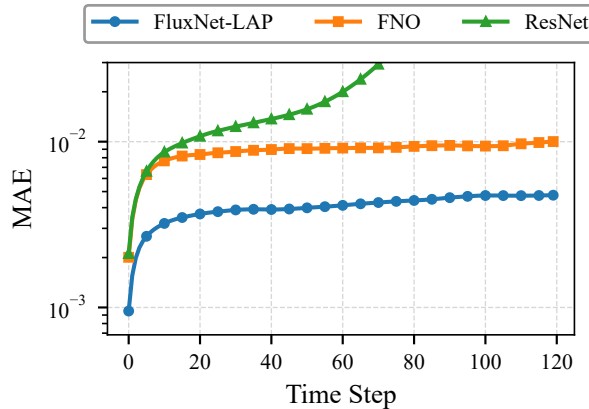

*Figure 1.* Rollout MAE over time for shallow water equations. MAE as a function of rollout time step for FluxNet-LAP, FNO, and ResNet from a single representative random seed on the shallow water equation test set.

These equations are widely used to model geophysical flows such as dam breaks, tsunami propagation, and coastal flooding. The key physical constraint is nonnegative depth ($h \geq 0$); negative water depths are physically meaningless and cause catastrophic instability through the velocity computation $\mathbf{v} = \mathbf{m}/h$.

Figure 1 reports rollout MAE over time. FluxNet-LAP maintains the lowest error throughout the rollout horizon, while ResNet-AR rapidly accumulates errors and becomes unstable. FNO-AR shows moderate error growth. Visualizations in Figures 6–8 of the appendix confirm that FluxNet-LAP closely matches the ground truth across all three fields, while ResNet-AR develops severe artifacts and FNO-AR produces distortions near wetting-drying boundaries, both driven by negative depth predictions that destabilize the learned dynamics.

Table 3 presents quantitative results at $T = 2$ ($2\times$ temporal extrapolation). With the ResNet backbone, FluxNet-LAP reduces depth MAE by 81% compared to the best projection baseline ($1.75$ vs. $9.41 \times 10^{-3}$ for Box+Mass). Conservation error reaches machine precision ($\sim 10^{-8}$), orders of magnitude lower than soft-constrained methods. Box+Mass projection achieves zero depth violations but incurs large momentum conservation errors ($\sim 10^1$) because the nonlinear correction interacts with the coupled multi-field structure, whereas FluxNet-LAP satisfies all constraints structurally without such artifacts.

To test modularity, we equip an FNO backbone with the LAP head (FluxNet-LAP (FNO) in Table 3), achieving the overall lowest depth MAE ($1.37 \times 10^{-3}$) and confirming that transport heads integrate with different encoder architectures. Ablation studies and sensitivity analysis of $\varepsilon$ are provided in Tables 7–8 of the appendix.

*Table 3.* Shallow water equations: baseline comparison at $T = 2$ ($2\times$ extrapolation). All methods use pushforward training. MAE values $\times 10^{-3}$. **Bold**: best in each backbone category.

| Method | $\mathcal{E}_{\text{MAE}}$ ($\times 10^{-3}$) | | | $\mathcal{E}_{\text{cons}}$ | | | $h$ Violation | |
| | $h$ | $m_x$ | $m_y$ | $h$ | $m_x$ | $m_y$ | $\mathcal{V}_{\text{lb}}$ (%) | $\mathcal{M}_{\text{lb}}$ |
|---|---|---|---|---|---|---|---|---|
| *ResNet backbone* | | | | | | | | |
| ResNet-AR | $72.8 \pm 84.3$ | $40.0 \pm 40.1$ | $58.9 \pm 75.8$ | 5.1e-2 | 1.5e+1 | 1.4e+1 | 3.15 | 1.4e-1 |
|   + SoftCons | $24.9 \pm 5.81$ | $30.9 \pm 5.56$ | $30.7 \pm 5.91$ | 1.9e-3 | 2.3e+0 | 1.8e+0 | 1.24 | 1.4e-2 |
|   + Box+Mass Proj. | $9.41 \pm 4.81$ | $21.2 \pm 21.4$ | $19.1 \pm 17.5$ | 3.2e-7 | 1.3e+1 | 1.7e+1 | **0.00** | – |
| **FluxNet-LAP** | $\mathbf{1.75 \pm 0.11}$ | $\mathbf{2.38 \pm 0.19}$ | $\mathbf{2.30 \pm 0.16}$ | 3.1e-8 | 6.1e-6 | 3.8e-6 | **0.00** | – |
| *FNO backbone* | | | | | | | | |
| FNO-AR | $5.93 \pm 0.20$ | $8.14 \pm 0.27$ | $8.05 \pm 0.28$ | 1.0e-2 | 6.2e+0 | 6.8e+0 | 0.80 | 2.9e-3 |
|   + SoftCons | $11.6 \pm 0.73$ | $16.3 \pm 1.11$ | $15.9 \pm 1.16$ | 2.0e-3 | 1.5e+0 | 1.6e+0 | 1.00 | 8.5e-3 |
|   + Box+Mass Proj. | $5.47 \pm 0.26$ | $8.33 \pm 0.50$ | $8.06 \pm 0.33$ | 3.1e-7 | 7.7e+0 | 5.9e+0 | **0.00** | – |
| **FluxNet-LAP (FNO)** | $\mathbf{1.37 \pm 0.12}$ | $\mathbf{1.81 \pm 0.19}$ | $\mathbf{1.89 \pm 0.11}$ | 3.1e-6 | 6.7e-6 | 3.5e-6 | **0.00** | – |

## 4.3. Traffic Flow

The 1D Lighthill–Whitham–Richards (LWR) traffic model (Lighthill & Whitham, 1955; Richards, 1956) requires density $\rho \in [0, 1]$, providing an ideal testbed for dual-bounded transport. We evaluate FluxNet-D under both periodic and Dirichlet boundary conditions, with the latter serving as the testbed for the ghost cell extension and the comparison setting for FluxGNN. Test cases include shocks, rarefaction waves, and complete flow blockages.

**Periodic boundary conditions.** Figure 2 shows density profiles at three time points on the periodic ring-road dataset. FluxNet-D accurately tracks shock formation and propagation while maintaining bounded predictions. FNO and ResNet baselines exhibit visible overshoot near discontinuities. Table 4 presents quantitative results. FluxNet-D achieves the best accuracy (MAE $2.79 \times 10^{-3}$) with machine-precision conservation. SigmoidBound eliminates violations but catastrophically degrades accuracy (MAE 82.7 vs. 2.79). FluxNet-D exhibits small violation rates with negligible conditional magnitudes (on the order of $10^{-3}$), demonstrating effective empirical dual-bound enforcement. FluxNet-D (FNO) confirms D-head modularity across architectures. DCL ablation is in the appendix.

**Dirichlet boundary conditions and comparison with flux-based surrogates.** To validate the ghost cell extension (Section 3.6) and compare against a flux-based conservative baseline, we construct a Dirichlet-boundary traffic flow dataset with three physically distinct inflow/outflow categories (details in the appendix). We compare FluxNet-D with FluxGNN (Horie & Mitsume, 2024) in its intended non-periodic regime, and also test FluxGNN-D (FluxGNN's backbone with our D-head) to assess modularity.

Table 5 (upper group) reports results at $\Delta t_{\text{model}} = 10\Delta t$. The periodic-only FluxNet-D fails catastrophically in the Dirichlet setting (MAE $717 \times 10^{-4}$), confirming that bound-

ary treatment is essential. FluxNet-D with Dirichlet ghost cells achieves MAE $11.7 \times 10^{-4}$, a $61\times$ improvement. FluxGNN achieves lower MAE ($3.54 \times 10^{-4}$), attributable to its more advanced GNN backbone; FluxNet uses a standard ResNet yet achieves reasonable accuracy. Notably, FluxGNN-D achieves the best MAE ($2.73 \times 10^{-4}$), demonstrating that our transport heads are modular: they plug into existing flux architectures for immediate improvement without altering the backbone. Rollout visualizations are provided in Figure 10 of the appendix.

**Large-timestep stability.** To test the CFL-free large-timestep capability (Remark 1), we compare FluxNet-D and FluxGNN at $\Delta t_{\text{model}} = 50\Delta t$, five times larger than the standard stride. Table 5 (lower group) shows that FluxNet-D maintains accurate rollouts (MAE $50.1 \times 10^{-4}$), while FluxGNN diverges (MAE $1150 \times 10^{-4}$). This validates the cumulative-transport paradigm: FluxGNN's flux rates inherit CFL constraints from explicit integration, while FluxNet's cumulative transport amounts enable stable prediction regardless of temporal stride. Rollout visualizations are provided in Figure 11 of the appendix.

## 4.4. Spinodal Decomposition

The 2D Cahn–Hilliard equation (Cahn & Hilliard, 1958; Chen, 2002) models spinodal decomposition in solid solutions, where an initially homogeneous alloy separates into two coexisting phases. The conserved concentration field $\phi \in [0, 1]$ represents solute distribution governed by a highly nonlinear chemical potential. This benchmark tests FluxNet-D's large-timestep capability, local transport behavior at coarse temporal resolution, and data efficiency through one-shot learning.

Since the local transport operator learned by FluxNet-D is spatiotemporally translation-invariant, the same local physics is encountered many times in a single simulation,

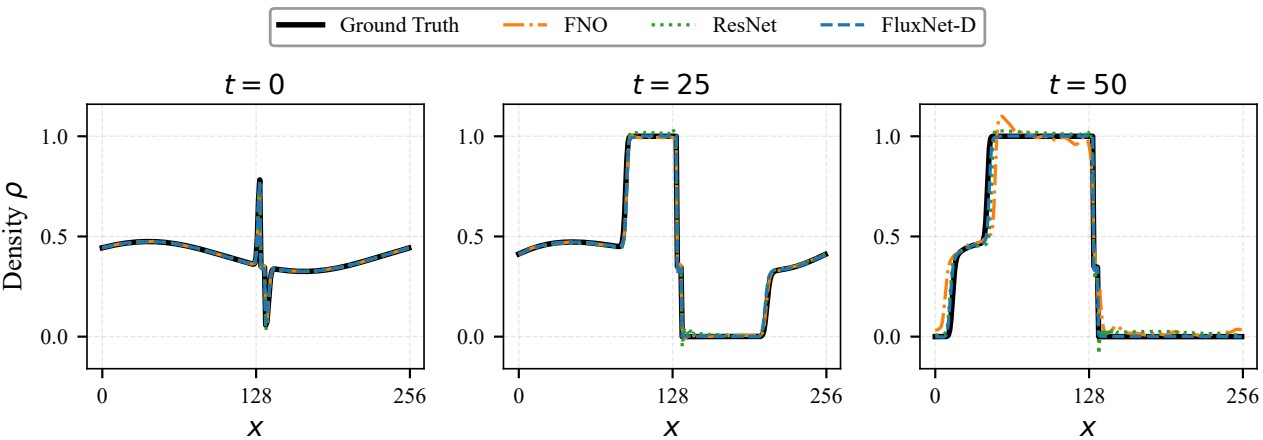

*Figure 2.* Traffic flow density profiles at selected time points. Comparison of density $\rho(x, t)$ predictions between ground truth, FluxNet-D, FNO, and ResNet at three time points ($t = 0$, $t = 25$, $t = 50$) on the periodic boundary condition dataset.

*Table 4.* Traffic flow (LWR model): baseline comparison at $T = 2$ with periodic boundary conditions. The density $\rho \in [0, 1]$ requires double-bound enforcement. MAE values $\times 10^{-3}$. **Bold**: best per backbone.

| Method | $\mathcal{E}_{\mathrm{MAE}}$ ($\times 10^{-3}$) | $\mathcal{E}_{\mathrm{cons}}$ | $\mathcal{V}_{\mathrm{lb}}$ (%) | $\mathcal{M}_{\mathrm{lb}}$ ($\times 10^{-3}$) | $\mathcal{V}_{\mathrm{ub}}$ (%) | $\mathcal{M}_{\mathrm{ub}}$ ($\times 10^{-3}$) |
|---|---|---|---|---|---|---|
| *ResNet backbone* | | | | | | |
| ResNet-AR | $8.09 \pm 0.91$ | 7.0e-3 | 2.33 | 7.20 | 0.78 | 6.88 |
| + SoftCons | $8.80 \pm 0.93$ | 4.8e-3 | 2.66 | 9.89 | 0.54 | 8.76 |
| + SigmoidBound + SoftCons | $82.7 \pm 18.5$ | 7.3e-2 | **0.00** | – | **0.00** | – |
| **FluxNet-D** | $\mathbf{2.79 \pm 0.79}$ | **6.7e-8** | 0.59 | **1.03** | 0.01 | **0.24** |
| *FNO backbone* | | | | | | |
| FNO-AR | $7.30 \pm 0.68$ | 2.7e-3 | 1.36 | 4.72 | **0.50** | 8.14 |
| + SoftCons | $8.40 \pm 0.89$ | 8.8e-4 | **1.11** | 8.57 | **0.50** | 14.5 |
| **FluxNet-D (FNO)** | $\mathbf{3.17 \pm 0.55}$ | **6.8e-8** | 2.47 | **2.15** | 1.02 | **4.07** |

*Table 5.* Traffic flow with Dirichlet boundary conditions. Upper group: four methods at $\Delta t_{\mathrm{model}} = 10\Delta t$. Lower group: FluxNet-D vs. FluxGNN at $\Delta t_{\mathrm{model}} = 50\Delta t$, testing large-timestep capability. MAE values $\times 10^{-4}$. **Bold**: best in each group.

| Method | $\Delta t_{\mathrm{model}}$ | $\mathcal{E}_{\mathrm{MAE}}$ ($\times 10^{-4}$) |
|---|---|---|
| FluxNet-D (Periodic) | $10\Delta t$ | $717 \pm 20.8$ |
| FluxNet-D (Dirichlet) | $10\Delta t$ | $11.7 \pm 0.23$ |
| FluxGNN | $10\Delta t$ | $3.54 \pm 1.08$ |
| **FluxGNN-D** | $10\Delta t$ | $\mathbf{2.73 \pm 0.95}$ |
| **FluxNet-D (Dirichlet)** | $50\Delta t$ | $\mathbf{50.1 \pm 4.85}$ |
| FluxGNN | $50\Delta t$ | $1150 \pm 87.9$ |

which therefore suffices as a rich training set (Jiao et al., 2025; Lan et al., 2025). We therefore adopt a one-shot learning setup: all models are trained from a single trajectory, with snapshots saved every $10\Delta t$ from $t = 2000\Delta t$ to $t = 52000\Delta t$.

We train three FluxNet-D models with temporal strides $10\Delta t$, $100\Delta t$, and $1000\Delta t$, progressively enlarging the

transport neighborhood radius ($R = 1, 2, 4$) to accommodate longer-range solute transport at larger time steps while maintaining full spatial resolution. Details of the training procedure are provided in the appendix.

Table 6 reports model configurations and performance. All three models achieve MAE in the $10^{-2}$ range, with the $100\Delta t$ model achieving the best accuracy ($2.16 \times 10^{-2}$). Phase field snapshots (Figure 12 in the appendix) show that the $100\Delta t$ model produces nearly indistinguishable predictions from the ground truth, while the other models show localized discrepancies due to chaotic sensitivity of coarsening dynamics.

Pointwise MAE alone cannot capture whether surrogates preserve the statistical properties that determine material performance. We evaluate the radial two-point correlation function $\bar{S}_2(r)$ (Torquato et al., 2002), a standard microstructure descriptor. Figure 3 shows the two-point statistics error over time; all three models maintain errors comparable to the intrinsic variability between two independent simula-

*Table 6.* Spinodal decomposition: multi-timestep FluxNet-D models. $R$: transport neighborhood radius; RF: theoretical receptive field; ERF: effective receptive field. Results at $T = 2$.

| $\Delta t_{model}$ | $R$ | $RF_{theo}$ | ERF | Speedup | $\mathcal{E}_{MAE}$ ($\times 10^{-2}$) |
|---|---|---|---|---|---|
| $10\Delta t$ | 1 | $19^2$ | $9.7^2$ | $0.55\times$ | 2.76 |
| $100\Delta t$ | 2 | $37^2$ | $12.2^2$ | $3.8\times$ | **2.16** |
| $1000\Delta t$ | 4 | $79^2$ | $19.0^2$ | $17.3\times$ | 8.39 |

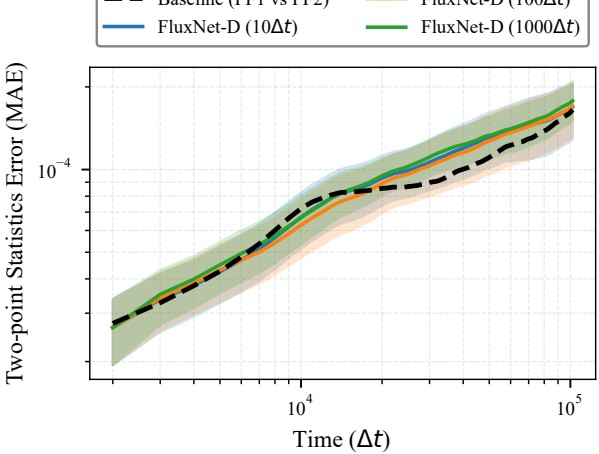

*Figure 3.* Two-point statistics error evolution for spinodal decomposition. MAE of the radial two-point correlation function as a function of simulation time for FluxNet-D models trained with different time step sizes. Shaded regions indicate $\pm 1$ standard deviation over 100 independent rollouts with different random initial conditions. The baseline error between two independent phase-field simulations is shown for reference.

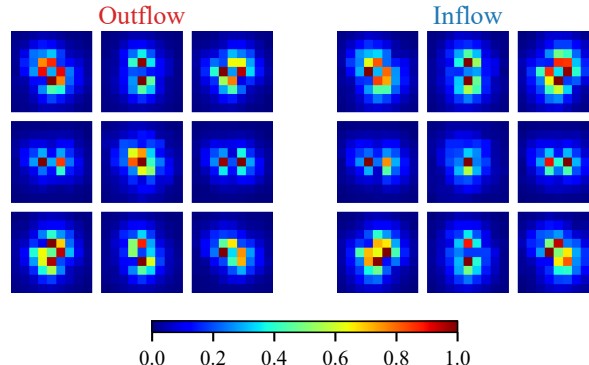

*Figure 4.* Effective receptive field (ERF) of FluxNet-D ($10\Delta t$). Left: outflow branch; Right: inflow branch. For each branch, the $K$ directional channels ($\hat{\pi}_{i,\mathbf{d}}$ or $\hat{\rho}_{i,\mathbf{d}}$) are spatially arranged by their offset $\mathbf{d} \in \mathcal{D}$, with the fraction channel ($\hat{\alpha}_i$ or $\hat{\beta}_i$) at center. The localized, anisotropic patterns confirm that learned transport operators depend only on local neighborhood information.

tions throughout the $2\times$ extrapolation regime, demonstrating preservation of coarsening statistics.

Wall-clock speedup over a GPU-accelerated phase-field solver (same NVIDIA A800 GPU) increases with temporal stride: $0.55\times$ at $10\Delta t$, $3.8\times$ at $100\Delta t$, and $17.3\times$ at $1000\Delta t$ (Table 6 and Figure 17 in the appendix). This progression highlights the practical importance of the CFL-free property: acceleration is achieved simply by enlarging the temporal stride, a strategy unavailable to CFL-constrained surrogates.

To verify that FluxNet-D operates as a local transport operator, we perform gradient-based effective receptive field (ERF) analysis (Luo et al., 2016). Figure 4 shows ERF patterns for the $10\Delta t$ model: directional channels exhibit anisotropy aligned with their transport direction. ERF sizes remain substantially smaller than theoretical maxima across all models (Table 6), confirming that the network learns genuinely local transport. Additional ERF visualizations are in the appendix.

## 5. Conclusion

We presented FluxNet, a neural PDE surrogate that learns capacity-constrained cumulative transport amounts rather than next-state values or instantaneous flux rates. The transport update structure guarantees exact discrete conservation at machine precision; modular transport heads (L, U, D) enforce physical bounds structurally as plug-in components compatible with different encoder architectures; and the absence of temporal integration eliminates CFL constraints, enabling large-timestep prediction at full spatial resolution via configurable transport neighborhoods. A ghost cell treatment extends the framework to non-periodic boundaries. Experiments on four benchmarks demonstrate machine-precision conservation, effective bound preservation, and superior stability compared to flux-rate surrogates at large time steps.

Several limitations suggest future directions. The D-head provides strong empirical dual-bound satisfaction, but not strict theoretical guarantees; structural preservation remains unresolved. The transport neighborhood size must be selected per temporal stride; adaptive neighborhoods could improve flexibility. Our fixed stencil structure does not preserve resolution invariance of spectral architectures such as FNO; developing resolution-independent transport parameterizations remains open. Extension to unstructured meshes is straightforward given compatibility with message-passing architectures (Horie & Mitsume, 2024). We consider only source-free conservation laws here; for cases involving source terms, they can be naturally handled via operator splitting. Non-periodic validation currently covers 1D Dirichlet conditions; extension to 2D and mixed boundary types follows from the same ghost cell mechanism but awaits validation.

## Acknowledgements

This work was supported by the National Natural Science Foundation of China (Grant No. 52471017) and the Advanced Materials-National Science and Technology Major Project (Grant No. 2025ZD0618501). We would like to thank Yu Chen, Yue Li, Weilong Ma, and Xiangju Liang for their helpful discussions. We also gratefully acknowledge the anonymous reviewers for their constructive comments and suggestions, which have significantly improved the quality of this paper.

## Impact Statement

This paper presents work whose goal is to advance scientific machine learning by introducing physically rigorous neural surrogates. FluxNet can accelerate complex PDE simulations by orders of magnitude without sacrificing physical validity, potentially reducing the computational energy footprint associated with large-scale scientific modeling in climate science and engineering. By structurally enforcing conservation laws and bounds, the method mitigates the risk of unphysical predictions that typically affect data-driven solvers, enhancing reliability in safety-critical applications. There are no specific negative societal consequences or dual-use risks that we feel must be highlighted here.

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

# A. Theoretical Properties and Proofs

This section provides complete proofs for the theoretical propositions stated in the main text, along with a detailed discussion of the guarantees and limitations of the D-head for dual-bounded transport.

## A.1. Proof of Proposition 1 (Discrete Conservation)

**Proposition 1.** Consider a regular grid $\Omega = \{1, 2, \ldots, N\}^d$ with periodic boundary conditions. Let $\mathcal{N}(i)$ denote the neighborhood of cell $i$ such that $j \in \mathcal{N}(i) \Leftrightarrow i \in \mathcal{N}(j)$ (symmetric stencil). For any transport field $\{F_{i \to j}\}_{i,j}$, the transport update

$$u_i^{t+1} = u_i^t - \sum_{j \in \mathcal{N}(i)} F_{i \to j} + \sum_{j \in \mathcal{N}(i)} F_{j \to i} \tag{7}$$

exactly preserves the global sum: $\sum_{i \in \Omega} u_i^{t+1} = \sum_{i \in \Omega} u_i^t$.

**Proof.** Summing the update equation over all cells:

$$\sum_{i \in \Omega} u_i^{t+1} = \sum_{i \in \Omega} u_i^t - \sum_{i \in \Omega} \sum_{j \in \mathcal{N}(i)} F_{i \to j} + \sum_{i \in \Omega} \sum_{j \in \mathcal{N}(i)} F_{j \to i}. \tag{8}$$

Consider the outflow term $S_{\text{out}} = \sum_{i \in \Omega} \sum_{j \in \mathcal{N}(i)} F_{i \to j}$. Each directed pair $(i, j)$ with $j \in \mathcal{N}(i)$ contributes $F_{i \to j}$ exactly once. Now consider the inflow term $S_{\text{in}} = \sum_{i \in \Omega} \sum_{j \in \mathcal{N}(i)} F_{j \to i}$. By the symmetry condition $j \in \mathcal{N}(i) \Leftrightarrow i \in \mathcal{N}(j)$, setting $i' = j$ and $j' = i$ gives:

$$S_{\text{in}} = \sum_{i \in \Omega} \sum_{j \in \mathcal{N}(i)} F_{j \to i} = \sum_{j' \in \Omega} \sum_{i' \in \mathcal{N}(j')} F_{i' \to j'} = S_{\text{out}}. \tag{9}$$

The symmetric neighborhood under periodic boundaries ensures that the set of all directed pairs $(i, j)$ is identical when enumerated from either endpoint. The outflow and inflow sums cancel:

$$\sum_{i \in \Omega} u_i^{t+1} = \sum_{i \in \Omega} u_i^t - S_{\text{out}} + S_{\text{in}} = \sum_{i \in \Omega} u_i^t. \quad \square \tag{10}$$

This proof holds for any transport values, including signed transport (N-head), nonnegative transport (P-head), or capacity-constrained transport (L/U/D-heads). Conservation is a structural property of the update rule itself, independent of how the transport amounts are computed by the neural network.

## A.2. Proofs of Propositions 2 and 3 (L-Head Lower Bound and U-Head Upper Bound)

**Proposition 2 (L-Head Lower Bound).** Let $\ell$ be a lower bound. Suppose $u_i^t \geq \ell$ for all $i \in \Omega$. Define the available amount $a_i = u_i^t - \ell \geq 0$. The L-head produces $K+1$ raw output channels from the backbone features $\mathbf{z}$, split into a scalar field $\hat{\alpha}_i$ and $K$ directional fields $\{\hat{\pi}_{i,\mathbf{d}}\}_{\mathbf{d} \in \mathcal{D}}$. An outflow fraction $\alpha_i = \sigma(\hat{\alpha}_i) \in (0, 1)$ is computed via sigmoid, and a distribution $\pi_{i \to (i+\mathbf{d})} = \text{softmax}_{\mathbf{d}}(\hat{\pi}_{i,\mathbf{d}}) \geq 0$ with $\sum_{\mathbf{d} \in \mathcal{D}} \pi_{i \to (i+\mathbf{d})} = 1$ is computed via softmax. The outgoing transport amount is $F_{i \to (i+\mathbf{d})} = a_i \cdot \alpha_i \cdot \pi_{i \to (i+\mathbf{d})}$. Then after one transport update, $u_i^{t+1} > \ell$ for all $i$.

**Proof.** The total outflow from cell $i$ is:

$$\text{Outflow}_i = \sum_{\mathbf{d} \in \mathcal{D}} F_{i \to (i+\mathbf{d})} = a_i \alpha_i \sum_{\mathbf{d} \in \mathcal{D}} \pi_{i \to (i+\mathbf{d})} = a_i \alpha_i. \tag{11}$$

The last equality follows from the softmax normalization $\sum_{\mathbf{d}} \pi_{i \to (i+\mathbf{d})} = 1$. Since $\alpha_i \in (0, 1)$ (the sigmoid function never saturates to exactly 0 or 1 for finite inputs), we have:

$$\text{Outflow}_i = a_i \alpha_i < a_i = u_i^t - \ell. \tag{12}$$

The total inflow to cell $i$ from its neighbors is:

$$\text{Inflow}_i = \sum_{\mathbf{d} \in \mathcal{D}} F_{(i-\mathbf{d}) \to i} = \sum_{\mathbf{d} \in \mathcal{D}} a_{i-\mathbf{d}} \alpha_{i-\mathbf{d}} \pi_{(i-\mathbf{d}) \to i} \geq 0, \tag{13}$$

where nonnegativity follows from $a_{i-\mathbf{d}} \geq 0$, $\alpha_{i-\mathbf{d}} > 0$, and $\pi_{(i-\mathbf{d}) \to i} \geq 0$. Combining:

$$u_i^{t+1} = u_i^t - \text{Outflow}_i + \text{Inflow}_i > u_i^t - (u_i^t - \ell) + 0 = \ell. \quad \square \tag{14}$$

The strict inequality $u_i^{t+1} > \ell$ (rather than $\geq$) arises because sigmoid outputs lie in the open interval $(0, 1)$. This provides a small numerical buffer above the bound, which is beneficial for stability in practice.

**Proposition 3 (U-Head Upper Bound).** The U-head is dual to the L-head. It produces $K+1$ raw output channels split into a scalar field $\hat{\beta}_i$ and $K$ directional fields $\{\hat{\rho}_{i,\mathbf{d}}\}_{\mathbf{d} \in \mathcal{D}}$. An inflow fraction $\beta_i = \sigma(\hat{\beta}_i) \in (0, 1)$ and a distribution $\rho_{i,\mathbf{d}} = \text{softmax}_{\mathbf{d}}(\hat{\rho}_{i,\mathbf{d}})$ allocate the remaining capacity $b_i = u_{\max} - u_i$ among neighbor directions. The incoming transport from neighbor $i + \mathbf{d}$ to cell $i$ is $F_{(i+\mathbf{d}) \to i} = b_i \cdot \beta_i \cdot \rho_{i,\mathbf{d}}$. By an analogous argument, total inflow to cell $i$ satisfies $\text{Inflow}_i = b_i \beta_i < b_i = u_{\max} - u_i$, and all outflow terms are nonnegative. Hence $u_i^{t+1} = u_i^t - \text{Outflow}_i + \text{Inflow}_i < u_i^t + b_i = u_{\max}$. The proof proceeds identically to Proposition 2 with sender and receiver roles exchanged.

### A.3. D-Head: Guarantees and Limitations

The D-head is designed for fields with dual bounds $u \in [\ell, u_{\max}]$, where a single one-sided capacity constraint is insufficient. The D-head produces $2(K+1)$ raw output channels, partitioned into two groups of $K+1$ channels for an outflow branch and an inflow branch.

The outflow branch computes $F_{i \to j}^{\text{out}}$ using the available amount $a_i = u_i - \ell$ following the L-head structure. By the same argument as Proposition 2, if the state update were performed using only the outflow branch, then $u_i^{t+1,\text{out}} > \ell$ would be guaranteed.

The inflow branch computes $F_{(i+\mathbf{d}) \to i}^{\text{in}}$ using the remaining capacity $b_i = u_{\max} - u_i$ following the U-head structure. By the dual argument, if the state update were performed using only the inflow branch, then $u_i^{t+1,\text{in}} < u_{\max}$ would be guaranteed.

Each branch individually is conservative (by Proposition 1) and satisfies a single-sided hard bound. However, the final state update averages the two change estimates: $u_i^{t+1} = u_i^t + \frac{1}{2}(\Delta u_i^{\text{out}} + \Delta u_i^{\text{in}})$. This averaged update does not inherit strict guarantees for both bounds unless $\Delta u^{\text{out}} = \Delta u^{\text{in}}$. Consider a scenario where the outflow branch prescribes large transport from cell $i$ to cell $j$ (the sender has ample available amount) but the inflow branch prescribes small transport (the receiver has limited capacity). The averaged transport may exceed what the receiver can safely absorb, leading to a potential upper-bound violation.

The Dual Consistency Loss (DCL) encourages agreement between the two branches during training (see Eq. (4) in the main text). When the two branches agree closely, the averaged update inherits the feasibility properties of both. In all experiments, we observe that DCL training drives violation rates below 3% with conditional violation magnitudes of $O(10^{-3})$, far smaller than unconstrained baselines. We report these violation statistics transparently as first-class metrics rather than claiming strict theoretical guarantees for dual-bounded enforcement.

## B. Convection–Diffusion

The 1D convection–diffusion equation serves as a minimal benchmark for comparing the three transport head variants (N, P, L), demonstrating that capacity-constrained heads outperform the unconstrained N-head.

### B.1. Governing Equation and Dataset

The governing equation describes the transport and diffusion of a conserved concentration field $c(x,t) \geq 0$:

$$\frac{\partial c}{\partial t} + u \frac{\partial c}{\partial x} = D \frac{\partial^2 c}{\partial x^2}, \quad (t,x) \in (0,T] \times [0,L], \tag{15}$$

with periodic boundary conditions $c(t,0) = c(t,L)$. The advection velocity $u$ governs the transport rate while the diffusion coefficient $D$ controls the smoothing rate. Total mass $\int_0^L c \, dx$ is conserved, and physical concentrations must remain nonnegative.

The dataset is constructed as follows. We set domain length $L = 1.0$ and diffusion coefficient $D = 0.005$. For each trajectory, the advection velocity is sampled uniformly as $u \sim \mathcal{U}(0.0, 0.2)$. Initial conditions are generated as superpositions

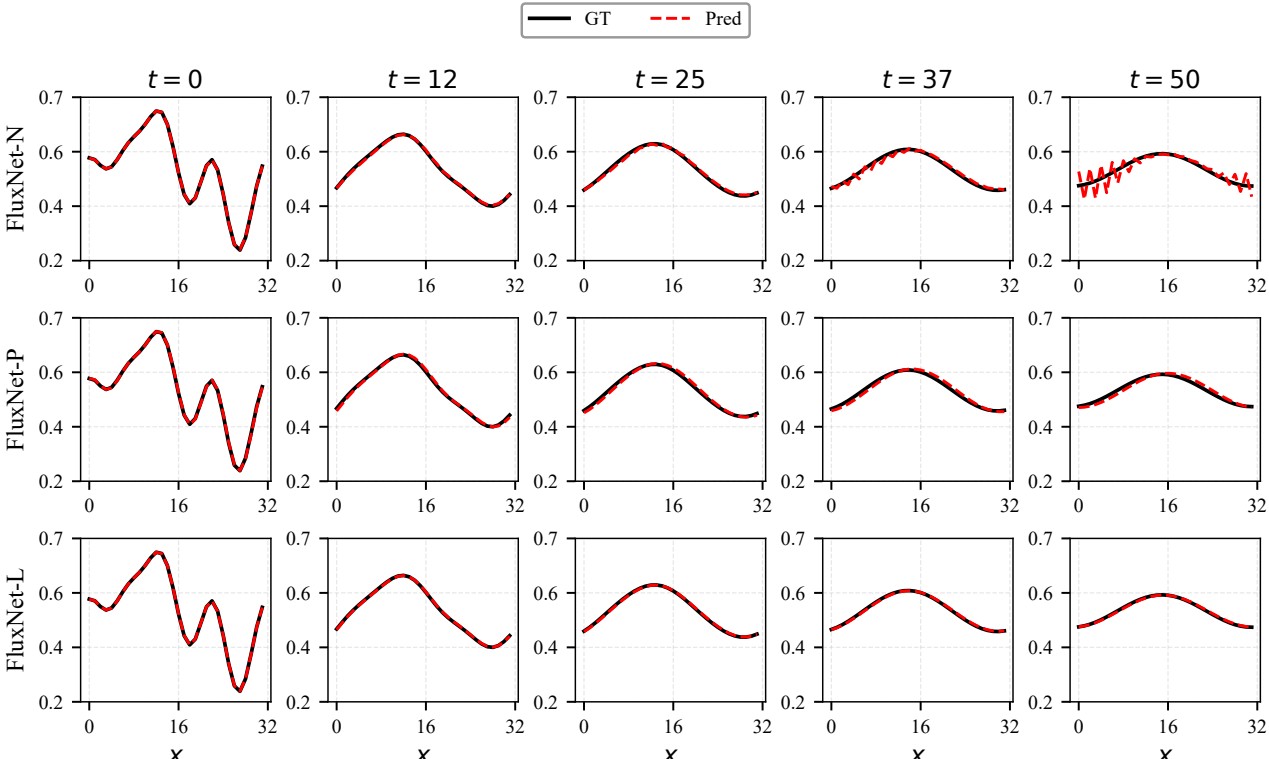

*Figure 5.* Concentration profiles at five time points comparing FluxNet-N (unconstrained), FluxNet-P (positive transport), and FluxNet-L (lower-bounded) variants on the 1D convection–diffusion equation. The N-head develops oscillations in later rollout stages, while P-head and L-head remain stable.

of four sinusoidal modes with random phases, with amplitudes scaled to ensure $c_0(x) \in [0, 1]$. The spatial domain is discretized on a grid of $N = 32$ points (downsampled from 64). The solver employs a Fourier spectral method with an exact integrating factor for temporal integration. We simulate $T_{\max} = 5.0$ time units, saving snapshots every $\Delta t_{\mathrm{save}} = 0.1$, yielding 51 frames per trajectory. The dataset split consists of 100 training, 10 validation, and 10 test trajectories.

### B.2. Model Configuration

All FluxNet models for this benchmark use a ResNet backbone with 16 base channels, 4 residual blocks, and kernel size 3. The transport neighborhood is a 3-point stencil (left and right neighbors, $R = 1$). Training employs the AdamW optimizer with an initial learning rate of $10^{-3}$ and weight decay of $10^{-2}$ for 300 epochs. A ReduceLROnPlateau scheduling strategy is adopted, halving the learning rate when the validation loss plateaus for 15 epochs. Since this benchmark focuses on comparing head variants rather than long-horizon stability, we use one-step training (no pushforward unrolling).

### B.3. Results

Figure 5 compares concentration profiles predicted by the three FluxNet variants against ground truth at five representative time points. At early rollout stages, all three variants show good agreement with the ground truth. As the rollout advances, their behaviors diverge: the L-head and P-head preserve high accuracy with virtually no deviation, whereas the N-head develops oscillations in later stages. The N-head permits arbitrary signed transport, allowing simultaneous opposing flows in both directions that create cancellation artifacts; these artifacts accumulate during rollout and occasionally lead to catastrophic error growth, as reflected in the large variance across random seeds ($16.73 \pm 24.22 \times 10^{-3}$ in Table 2). The P-head eliminates this failure mode by constraining all transport to be nonnegative via softplus, effectively restricting the output space and preventing sign oscillations. The L-head further restricts the output space by limiting total outflow to the available amount above zero, achieving a slight accuracy improvement over the P-head. Both constrained heads exhibit low variance across seeds, confirming stable training. All variants maintain conservation error at machine precision.

## C. Shallow Water Equations

The 2D shallow water equations provide a challenging benchmark for coupled multi-field conservation with a strict positivity constraint on water depth.

### C.1. Governing Equations and Dataset

The governing equations couple three conserved fields: water depth $h \geq 0$ and momentum components $(m_x, m_y) = h(u, v)$ where $(u, v)$ is the velocity field:

$$\frac{\partial h}{\partial t} + \nabla \cdot (h\mathbf{v}) = 0, \tag{16}$$

$$\frac{\partial \mathbf{m}}{\partial t} + \nabla \cdot (\mathbf{m} \otimes \mathbf{v}) = -gh\nabla h, \tag{17}$$

where $g = 9.81$ m/s$^2$ is gravitational acceleration. The physical constraint $h \geq 0$ is strict; negative water depths are physically meaningless and cause numerical instabilities through the velocity computation $\mathbf{v} = \mathbf{m}/h$.

The dataset is constructed on a doubly-periodic domain $[0, 10]^2$ discretized on a $64 \times 64$ grid (downsampled from $128 \times 128$). The numerical solver employs a finite volume method with Rusanov flux and SSP-RK3 time stepping at $\Delta t = 0.004$. All initial condition categories contain a fraction of dry cells (zero depth) ranging from 5% to 35% of the domain. The four categories are: Case A1 (Gaussian superposition, zero momentum), Case A2 (Fourier synthesis, zero momentum), Case B1 and B2 (extending A1 and A2 with nonzero momentum fields). The training horizon is $T_{\text{train}} = 2.4$, and test rollouts extend to $T_{\text{test}} = 4.8$ ($2\times$ extrapolation). The dataset split consists of 50 training, 20 validation, and 50 test trajectories, stratified across the four types.

### C.2. Model Configuration

FluxNet-LAP uses a ResNet backbone with 64 base channels, 6 residual blocks, and kernel size 5. The transport neighborhood is a $3 \times 3$ stencil (8 neighbors, $R = 1$). Training employs the AdamW optimizer with learning rate $10^{-3}$, weight decay $10^{-2}$, and pushforward unrolling with a fixed unroll length of 5 steps over 300 epochs. The learning rate is scheduled via ReduceLROnPlateau (factor 0.5, patience 15 epochs).

### C.3. Qualitative Results

Figures 6, 7, and 8 show the time evolution of the three conserved fields ($h$, $m_x$, $m_y$) comparing ground truth against FluxNet-LAP, FNO, and ResNet predictions at six time steps. FluxNet-LAP maintains accurate field evolution throughout the rollout horizon, closely tracking ground truth wetting-drying fronts and momentum transport patterns. The ResNet baseline develops severe checkerboard artifacts by mid-trajectory, visible across all three fields, indicating instability caused by negative depth predictions that propagate and destabilize the learned dynamics during autoregressive rollout. FNO exhibits localized distortions near wave fronts and wetting-drying boundaries, though it maintains better global structure than ResNet. These qualitative observations are consistent with the quantitative results in Table 3 of the main text.

### C.4. Ablation Study

Table 7 presents the ablation study results. Enforcing $h \geq 0$ via the L-head is essential: the PPP variant (P-head for all fields) exhibits a 3.13% depth violation rate and approximately $67\times$ higher depth MAE than the full LAP model, with large variance across seeds indicating unstable training. The LPP variant (L-head for depth, P-head for momentum without advection-pressure decomposition) achieves zero depth violations but shows higher momentum error than LAP, indicating that the advection-pressure decomposition improves momentum transport learning. The PAP variant (P-head for depth with advection-pressure coupling for momentum) diverges during training because the momentum advection term $F_{i \to j}^{m,\text{adv}} = F_{i \to j}^h \cdot m_i/(h_i + \varepsilon)$ becomes unstable when the P-head permits negative depth predictions. Removing the $h^2$ pressure gating increases momentum error by approximately 35–40%, confirming that the gating improves robustness near dry regions where spurious momentum exchange should be suppressed. Pushforward training provides a clear benefit for this benchmark: including it reduces depth MAE from 2.73 to $1.75 \times 10^{-3}$ (a 36% improvement) and momentum MAE by 32–35%, confirming its value for stabilizing long-horizon coupled dynamics.

*Table 7.* Shallow water equations: ablation study at $T = 2$. PPP: P-head for all fields; LPP: L-head for $h$, P-head for momentum; PAP: P-head for $h$ with advection-pressure momentum decomposition. All MAE values are $\times 10^{-3}$. **Bold**: best per column among non-diverged variants.

| | $\mathcal{E}_{\text{MAE}}$ $(\times 10^{-3})$ | | | $\mathcal{E}_{\text{cons}}$ | | | $h$ Violation |
|---|---|---|---|---|---|---|---|
| Variant | $h$ | $m_x$ | $m_y$ | $h$ | $m_x$ | $m_y$ | $\mathcal{V}_{\text{lb}}$ (%) |
| FluxNet-PPP | $117 \pm 155$ | $106 \pm 99.1$ | $94.8 \pm 86.2$ | 1.55e-7 | 1.1e-4 | 1.1e-4 | 3.13 |
| FluxNet-LPP | $3.89 \pm 0.92$ | $7.86 \pm 4.49$ | $18.8 \pm 18.1$ | 3.27e-8 | 1.2e-4 | 1.1e-4 | **0.00** |
| FluxNet-PAP | | | Diverged | | | | |
| LAP (w/o pressure gating) | $2.46 \pm 0.27$ | $3.20 \pm 0.35$ | $3.28 \pm 0.27$ | 3.27e-8 | 1.1e-4 | 1.3e-4 | 0.00 |
| LAP (w/o pushforward) | $2.73 \pm 0.32$ | $3.52 \pm 0.40$ | $3.34 \pm 0.36$ | 3.28e-8 | **5.9e-6** | **3.5e-6** | 0.00 |
| FluxNet-LAP | $\mathbf{1.75 \pm 0.11}$ | $\mathbf{2.38 \pm 0.19}$ | $\mathbf{2.30 \pm 0.16}$ | **3.14e-8** | 6.1e-6 | 3.8e-6 | 0.00 |

*Table 8.* Sensitivity of FluxNet-LAP to the regularization parameter $\varepsilon$ in the momentum advection term $F_{i \to j}^{m,\text{adv}} = F_{i \to j}^{h} \cdot m_i / (h_i + \varepsilon)$. MAE converges for $\varepsilon \le 10^{-4}$ and remains stable through $10^{-8}$.

| $\varepsilon$ | $\mathcal{E}_{\text{MAE}}$ |
|---|---|
| $10^{-2}$ | $5.53 \times 10^{-3}$ |
| $10^{-3}$ | $4.77 \times 10^{-3}$ |
| $10^{-4}$ | $4.76 \times 10^{-3}$ |
| $10^{-5}$ | $4.76 \times 10^{-3}$ |
| $\mathbf{10^{-6}}$ **(Ours)** | $4.76 \times 10^{-3}$ |
| $10^{-7}$ | $4.76 \times 10^{-3}$ |
| $10^{-8}$ | $4.76 \times 10^{-3}$ |

### C.5. Sensitivity to $\varepsilon$

Table 8 reports the sensitivity of FluxNet-LAP to $\varepsilon$. MAE decreases from $5.53 \times 10^{-3}$ at $\varepsilon = 10^{-2}$ and converges to $4.76 \times 10^{-3}$ for $\varepsilon \le 10^{-4}$, remaining stable through $10^{-8}$. The insensitivity for $\varepsilon \le 10^{-4}$ indicates robustness: $\varepsilon$ only matters in rare near-dry cells ($h_i \approx 0$), and any sufficiently small value yields equivalent performance. We use $\varepsilon = 10^{-6}$ as the default.

## D. Traffic Flow

The 1D traffic flow problem based on the Lighthill–Whitham–Richards (LWR) model provides an ideal testbed for dual-bounded transport, as the density must satisfy $\rho \in [0, 1]$ and the dynamics feature shocks and rarefaction waves.

### D.1. Governing Equation

The governing equation is a scalar conservation law with a nonlinear flux function:

$$\frac{\partial \rho}{\partial t} + \frac{\partial}{\partial x}[v_{\max}(x) \, \rho \, (1 - \rho)] = 0, \tag{18}$$

where $\rho \in [0, 1]$ represents the normalized traffic density (fraction of road capacity), and $v_{\max}(x)$ is the spatially varying maximum velocity. The flux function $Q(\rho) = v_{\max}\rho(1 - \rho)$ is concave with maximum at $\rho = 0.5$, leading to characteristic shock formation when high-density regions encounter low-density regions.

### D.2. Periodic Boundary Conditions

#### D.2.1. DATASET

The periodic dataset is constructed on a periodic domain $[0, 10)$ (ring road) discretized on $N = 256$ grid points without spatial downsampling. The numerical solver uses a first-order finite volume method with Rusanov flux at $\Delta t = 0.016$. Snapshots are saved every $10\Delta t$. Initial conditions span seven case types probing different physical regimes: Case 1 (traffic jam, ramp-plateau profile), Case 2A (speed limit zone), Case 2B (red light), Cases 3+, 3-, $3_0$ (Riemann problems with forward, backward, and stationary shocks), and Case 4 (rarefaction waves). The training horizon is $T_{\text{train}} = 4.0$, and test

*Table 9.* Traffic flow (periodic BC): ablation study at $T = 2$. P/L/U: single-constraint heads. The D-head with DCL achieves the best trade-off between accuracy and balanced bound enforcement.

| Variant | $\mathcal{E}_{\text{MAE}}$ ($\times 10^{-3}$) | $\mathcal{E}_{\text{cons}}$ ($\times 10^{-7}$) | $\mathcal{V}_{\text{lb}}$ (%) | $\mathcal{M}_{\text{lb}}$ ($\times 10^{-3}$) | $\mathcal{V}_{\text{ub}}$ (%) | $\mathcal{M}_{\text{ub}}$ ($\times 10^{-3}$) |
|---|---|---|---|---|---|---|
| FluxNet-P | $4.40 \pm 1.35$ | 5.30 | 2.72 | 3.95 | 0.86 | 3.66 |
| FluxNet-L | $2.68 \pm 0.75$ | 0.915 | **0.00** | – | 1.16 | 5.95 |
| FluxNet-U | $2.14 \pm 0.54$ | 1.25 | 2.53 | 2.75 | **0.00** | – |
| D (w/o DCL) | $\mathbf{1.52 \pm 0.14}$ | **0.667** | 2.26 | 0.83 | 0.41 | 1.18 |
| D (w/o pushforward) | $4.08 \pm 0.46$ | 0.671 | 0.01 | **0.24** | 0.43 | **0.46** |
| **FluxNet-D** | $2.79 \pm 0.79$ | 0.673 | 0.59 | **1.03** | 0.01 | **0.24** |

rollouts extend to $T_{\text{test}} = 8.0$ ($2\times$ extrapolation). The dataset split consists of 100 training, 50 validation, and 100 test trajectories.

### D.2.2. MODEL CONFIGURATION

FluxNet-D uses a ResNet backbone with 32 base channels, 6 residual blocks, and kernel size 5. The transport neighborhood is an 11-point stencil ($R = 5$). Training employs the AdamW optimizer with learning rate $10^{-3}$, weight decay $10^{-2}$, pushforward unrolling with a fixed unroll length of 5 steps, DCL weight $\gamma = 1.0$, and a ReduceLROnPlateau scheduler that halves the learning rate when the validation loss stagnates for 15 epochs. Training proceeds for 300 epochs.

### D.2.3. ABLATION STUDY

Table 9 presents the ablation study. Single-sided heads strictly enforce their respective bounds but violate the opposite: the L-head achieves zero lower-bound violations but 1.16% upper-bound violations, while the U-head shows 2.53% lower-bound violations but zero upper-bound violations. The P-head, which only enforces nonnegative transport without capacity constraints, exhibits violations on both sides. Removing DCL from the D-head yields the lowest MAE ($1.52 \times 10^{-3}$) but substantially increases violation rates on both bounds (lower-bound violations from 0.59% to 2.26%, upper-bound from 0.01% to 0.41%), confirming that dual consistency regularization trades some pointwise accuracy for balanced bound enforcement. Pushforward training is important for this shock-dominated benchmark: removing it increases MAE from 2.79 to 4.08 ($\times 10^{-3}$), a $1.5\times$ degradation, because shock dynamics are inherently sensitive to distributional shift during autoregressive rollout.

### D.3. Dirichlet Boundary Conditions

#### D.3.1. DATASET CONSTRUCTION

The Dirichlet traffic flow dataset is governed by $\partial_t \rho + \partial_x[\rho(1 - \rho)] = 0$ on the spatial domain $[0, L]$ with $L = 10$, subject to time-invariant Dirichlet conditions $\rho(0, t) = \rho_L$ and $\rho(L, t) = \rho_R$. The dataset comprises 150 samples organized into three physically distinct categories: **(D1)** Steady inflow-outflow ($\rho_L = \rho_R$), where initial perturbations relax toward a spatially uniform equilibrium; **(D2)** Shock formation ($\rho_L > \rho_R$), where a high-density left boundary drives a rightward-propagating shock; and **(D3)** Congestion back-propagation ($\rho_L < \rho_R$), where a downstream bottleneck generates a leftward-moving congestion wave. For each sample, the initial condition is a random superposition of 1–3 sinusoidal harmonics with randomized amplitudes and phases, intentionally mismatched with the prescribed boundary values to introduce boundary-layer dynamics.

Reference solutions are computed with the same first-order finite-volume scheme as the periodic case (Rusanov flux, 256 cells, $\Delta t = 0.016$). Snapshots are saved every $10\Delta t$ (or every $50\Delta t$ for the large-timestep experiments), without spatial downsampling. The two boundary values are appended to each snapshot, yielding tensors that explicitly encode the Dirichlet conditions. The training horizon is $T_{\text{train}} = 4.0$ and test rollouts extend to $T_{\text{test}} = 8.0$ ($2\times$ extrapolation). The dataset is split into 50 training, 50 validation, and 50 test samples, balanced equally across the three categories.

### D.3.2. GHOST CELL IMPLEMENTATION

Figure 9 illustrates the ghost cell boundary treatment for FluxNet-D under Dirichlet conditions. Given the physical domain $\Omega$ of $N = 256$ cells with prescribed boundary densities $\rho_L = \rho(0, t)$ and $\rho_R = \rho(L, t)$, we pad $R$ ghost cells on each side (where $R$ is the stencil radius), filling them with the prescribed Dirichlet values. A binary identity channel (1 for interior cells, 0 for ghost cells) is appended to the input, enabling the network to learn boundary-aware transport prediction without modifying the core transport mechanism. The backbone uses replicate padding instead of circular padding. The transport update proceeds on the extended domain; only interior cell values are retained as the next state.

### D.3.3. MODEL CONFIGURATION

For the standard temporal stride ($\Delta t_{\mathrm{model}} = 10\Delta t$), FluxNet-D uses the same architecture hyperparameters as in the periodic case: a ResNet backbone with 32 base channels, 6 residual blocks, kernel size 5, and an 11-point transport stencil ($R = 5$). For the large temporal stride ($\Delta t_{\mathrm{model}} = 50\Delta t$), the transport neighborhood is widened to a 15-point stencil ($R = 7$) to accommodate the larger per-step displacement; all other hyperparameters remain unchanged. Training settings—AdamW with learning rate $10^{-3}$, weight decay $10^{-2}$, pushforward unrolling of 5 steps, DCL weight $\gamma = 1.0$, ReduceLROnPlateau scheduler (patience 15, factor 0.5), and 300 epochs—are identical to the periodic case for both temporal strides.

### D.3.4. FLUXGNN IMPLEMENTATION

FluxGNN (Horie & Mitsume, 2024) is a finite-volume-inspired graph neural network that predicts inter-cell flux rates and updates states via divergence-form integration, preserving conservation by construction. We implement FluxGNN on the regular 1D grid for the Dirichlet traffic flow benchmark. FluxGNN-D denotes the variant where FluxGNN's backbone is equipped with our D-head, replacing the direct flux-rate prediction with capacity-constrained transport parameterization. To ensure a fair comparison, we match FluxGNN's key hyperparameters (hidden dimensions, number of layers) to produce a model with comparable parameter count to FluxNet-D, and train all methods for the same number of epochs with the same optimizer settings.

### D.3.5. ROLLOUT VISUALIZATIONS

Figures 10 and 11 provide rollout visualizations. At $\Delta t_{\mathrm{model}} = 10\Delta t$ (Figure 10), the periodic-only FluxNet-D fails catastrophically, while all three Dirichlet-aware methods (FluxNet-D with ghost cells, FluxGNN, and FluxGNN-D) produce accurate rollouts. At $\Delta t_{\mathrm{model}} = 50\Delta t$ (Figure 11), FluxNet-D maintains accurate density tracking throughout the $2\times$ extrapolation horizon, while FluxGNN diverges, visually confirming the CFL limitation of the flux-rate approach and the CFL-free advantage of cumulative transport.

## E. Spinodal Decomposition

The 2D Cahn–Hilliard equation models spinodal decomposition in solid solutions, a classical process in materials microstructure evolution where an initially homogeneous alloy separates into two coexisting phases through uphill diffusion. The conserved concentration field $\phi \in [0, 1]$ represents solute distribution. This benchmark demonstrates FluxNet-D's large-timestep capability, its behavior as a local transport operator and data efficiency through one-shot learning.

### E.1. Governing Equation

The governing equation is a fourth-order parabolic PDE:

$$\frac{\partial \phi}{\partial t} = M\nabla^2 \mu, \quad \mu = f'(\phi) - \kappa\nabla^2\phi, \tag{19}$$

where $\phi \in [0, 1]$ is the concentration, $M = 1.0$ is the mobility coefficient, $\kappa = 0.357$ J/m is the gradient energy coefficient, and $\mu$ is the chemical potential. The bulk free energy density follows a regular solution model for Au-Pt alloy at temperature $T = 973.15$ K:

$$f(\phi) = \frac{RT}{v_m}[\phi\ln\phi + (1 - \phi)\ln(1 - \phi)] + \phi(1 - \phi)[A_0 + A_1(1 - 2\phi)], \tag{20}$$

with $A_0 = 15000 + 6.1T$ and $A_1 = -7600 + 3.55T$. The spinodal decomposition process proceeds through three distinct phases: initial noise relaxation ($0$–$20\Delta t$), rapid phase separation ($20$–$600\Delta t$), and slow coarsening ($> 600\Delta t$) where the characteristic domain size grows over time.

### E.2. Dataset and Training

The dataset is constructed on a periodic domain $[0, 128]^2$ discretized on a $128 \times 128$ grid for training and evaluation. The numerical solver uses explicit finite differences with $\Delta t = 0.01$. Initial conditions are generated as $\phi_0(\mathbf{x}) = \bar{\phi}_0 + 0.05 \times (\zeta(\mathbf{x}) - 0.5)$ where $\zeta(\mathbf{x}) \sim \mathcal{U}(0, 1)$ independently at each grid point and $\bar{\phi}_0 = 0.60$ is the mean composition.

We employ a one-shot learning paradigm exploiting the spatial self-similarity of spinodal decomposition: training uses a single long trajectory rather than many short ones. All three stride models share the same training set: a single trajectory evolved from $t = 2000\Delta t$ to $t = 52000\Delta t$, with snapshots saved every $10\Delta t$. An overlapping strategy is used to construct input-target pairs: for a model with stride $s\Delta t$, consecutive snapshots at $10\Delta t$ intervals provide pairs $(\phi^{k \cdot 10\Delta t}, \phi^{k \cdot 10\Delta t + s\Delta t})$ for all valid $k$. This yields a large number of overlapping training pairs for large timestep models even from a single trajectory. Test evaluation uses 20 trajectories with different random seeds, extended to $t = 102000\Delta t$ (corresponding to $2\times$ temporal extrapolation).

### E.3. Model Configurations

The three FluxNet-D models use ResNet backbones with 32 base channels but different configurations. The $10\Delta t$ model uses 4 residual blocks with kernel size 3 and a $3 \times 3$ transport neighborhood ($R = 1$). The $100\Delta t$ model uses 4 blocks with kernel size 5 and a $5 \times 5$ neighborhood ($R = 2$). The $1000\Delta t$ model uses 6 blocks with kernel size 7 and a $9 \times 9$ neighborhood ($R = 4$). All models are trained with the AdamW optimizer (learning rate $10^{-3}$, weight decay $10^{-2}$) for 100 epochs with DCL weight $\gamma = 1.0$. The learning rate is reduced by a factor of $0.5$ if no improvement in validation loss is observed for 15 consecutive epochs.

### E.4. Qualitative Results

Figure 12 compares concentration field snapshots at four key time points between ground truth and all three FluxNet-D models. All models produce visually realistic microstructures with appropriate domain morphology and characteristic length scales that grow over time consistent with coarsening dynamics. The $100\Delta t$ model shows the closest agreement with the ground truth; its absolute error map at $2T$ reveals only minor discrepancies concentrated near phase interfaces. The $10\Delta t$ and $1000\Delta t$ models exhibit larger localized errors in certain regions, reflecting the chaotic sensitivity of coarsening dynamics: small errors near phase interfaces can lead to divergent merging pathways (e.g., whether two adjacent domains merge or remain separate at a particular time).

### E.5. Statistical Evaluation

In materials science, the functional properties of a microstructure depend not on the exact spatial arrangement of phases but on their statistical characteristics. Evaluating whether a surrogate preserves these statistical properties is therefore essential for materials applications.

#### E.5.1. TWO-POINT CORRELATION

The two-point correlation function $S_2(\mathbf{r})$ quantifies the probability that two points separated by vector $\mathbf{r}$ belong to the same phase. It is computed via FFT-based autocorrelation as $S_2(\mathbf{r}) = \langle \phi(\mathbf{x})\phi(\mathbf{x} + \mathbf{r}) \rangle_{\mathbf{x}}$, and the radial average $\bar{S}_2(r)$ provides a one-dimensional summary capturing the characteristic length scale of the microstructure. The position of the first minimum of $\bar{S}_2(r)$ serves as a proxy for the mean domain size. We evaluate the MAE between predicted and reference radial correlations as a statistical accuracy metric complementing pointwise error. Figure 13 plots $\bar{S}_2(r)$ at four key times, comparing ground truth with all three models. The correlation functions show excellent agreement at all times. The characteristic length scale, indicated by the position of the first minimum, is correctly captured and grows appropriately with time.

#### E.5.2. PHASE VOLUME FRACTION

Figure 14 tracks the phase volume fractions over time, defined as the fraction of pixels with $\phi \geq 0.6$ (high-concentration phase) and $\phi < 0.6$ (low-concentration phase). All FluxNet-D models accurately reproduce the phase fraction evolution, with predicted curves closely tracking the ground truth throughout the $2\times$ extrapolation regime. This confirms that the conservation constraint maintained by the FluxNet architecture successfully preserves the global composition balance, translating to correct mass partitioning between phases at the macroscopic level.

### E.6. Effective Receptive Field Analysis

We perform gradient-based effective receptive field (ERF) analysis to verify that FluxNet-D operates as a local transport operator. For each output channel $c$ at spatial location $(y, x)$, we compute the gradient magnitude $G_c(y, x) = |\partial f_c(y, x)/\partial \mathbf{X}|$, where $f_c(y, x)$ is the channel output before the conservative update and $\mathbf{X}$ is the input concentration field. We sample 10 random images from the test set and 10 random spatial locations per image, compute gradient maps for all sampled locations, average across samples, and normalize by the maximum value. The effective receptive field is defined as pixels where the normalized gradient exceeds 1% of the maximum, and ERF size is computed as the square root of the pixel count. The theoretical receptive field for a backbone with initial convolution of kernel size $k$ followed by $B$ residual blocks (each with two $k \times k$ convolutions) is TRF $= k + 2B(k - 1)$.

Figures 15 and 16 visualize ERF patterns for the $100\Delta t$ and $1000\Delta t$ models respectively (the $10\Delta t$ model is shown in Figure 4 of the main text). Models trained with larger time steps exhibit proportionally larger effective receptive fields: approximately $9.7 \times 9.7$ pixels for $10\Delta t$, $12.2 \times 12.2$ for $100\Delta t$, and $19.0 \times 19.0$ for $1000\Delta t$. Importantly, all models maintain ERF sizes substantially smaller than their theoretical maxima: the ERF/RF utilization ratios are 26.0%, 10.8%, and 5.8% for the three models respectively. This confirms that FluxNet-D learns genuinely local transport operators rather than global convolutional mappings, supporting the physical interpretation that the learned dynamics represent local solute transport between neighboring cells.

### E.7. Computation Time Comparison

Figure 17 compares wall-clock time for completing $50{,}000\Delta t$ of simulation across domain sizes from $256^2$ to $4096^2$ grid points, all measured on the same NVIDIA A800 GPU. The GPU-accelerated phase-field solver time scales with domain size.

FluxNet-D inference time is determined by two factors: the number of forward passes (inversely proportional to the temporal stride) and the cost per pass (proportional to both the domain size and the number of model parameters). The $10\Delta t$ model requires 5,000 forward passes, the $100\Delta t$ model requires 500, and the $1000\Delta t$ model requires only 50. At the training resolution of $1024^2$, the $10\Delta t$ model is actually slower than the solver ($0.55\times$ speedup) because inference overhead per pass outweighs the savings from a modest stride reduction. The $100\Delta t$ model achieves $3.8\times$ speedup and the $1000\Delta t$ model achieves $17.3\times$ speedup. The efficiency advantage of the coarser-timestep models increases further at larger domain sizes.

This progression directly illustrates the practical consequence of the CFL-free property. Flux-rate-based surrogates that inherit CFL constraints are effectively limited to temporal strides comparable to the $10\Delta t$ model, where no meaningful speedup over optimized numerical solvers is achieved. FluxNet's ability to stably predict at $100\Delta t$ or $1000\Delta t$ without temporal integration converts the CFL-free property into concrete computational acceleration.

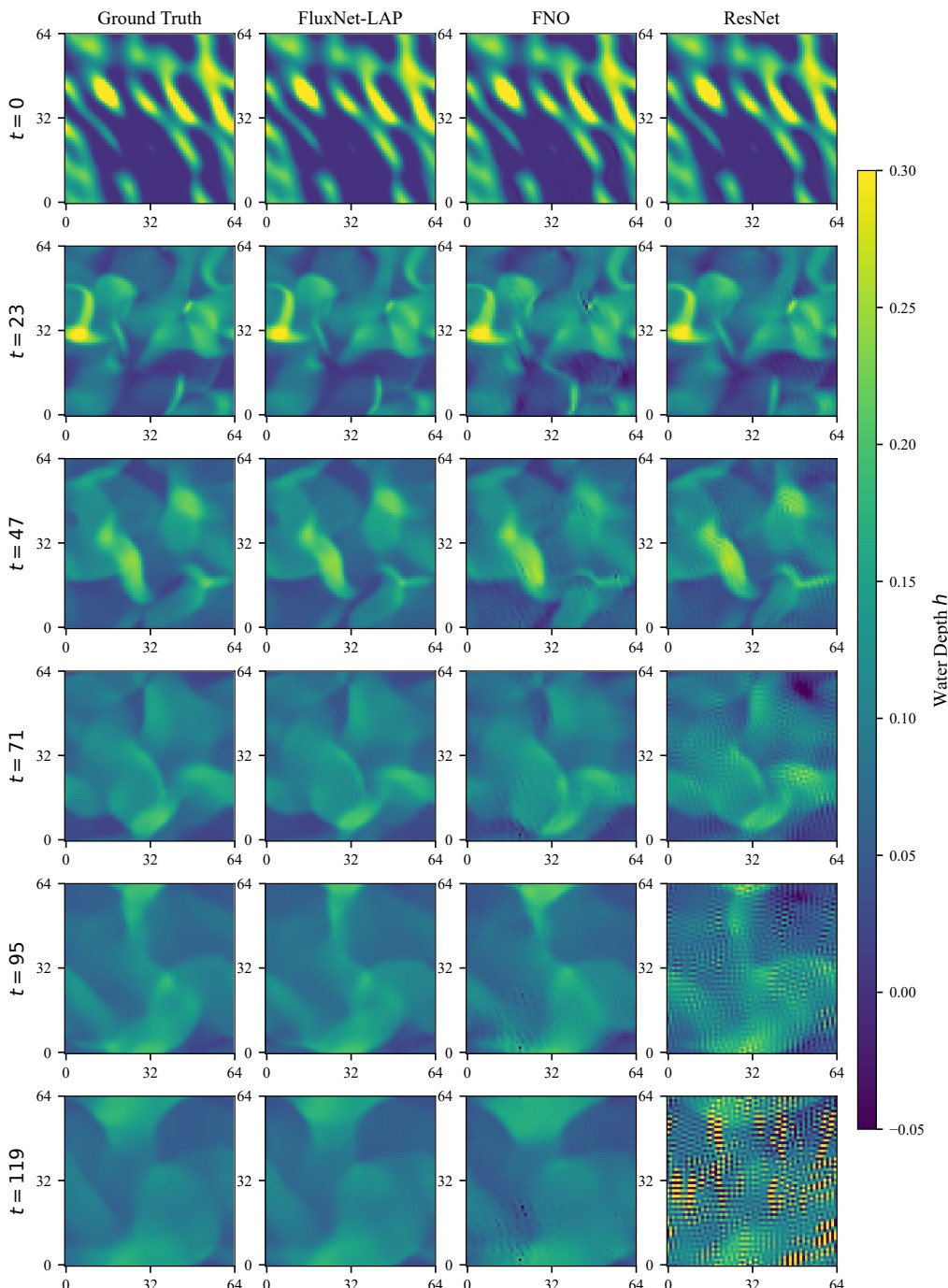

*Figure 6.* Time evolution of the water depth field $h(x, y, t)$ comparing ground truth, FluxNet-LAP, FNO, and ResNet at six time steps spanning the rollout horizon.

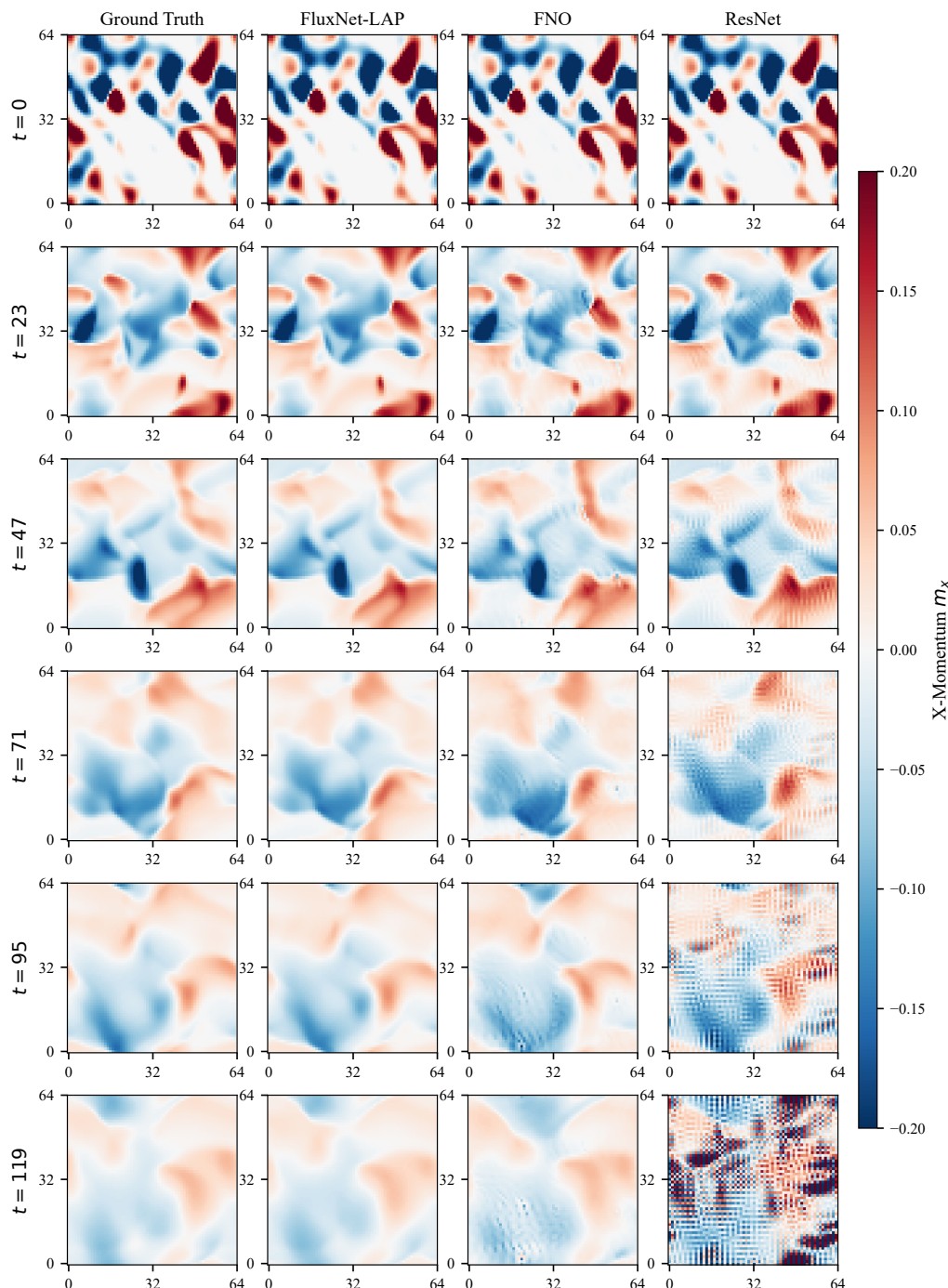

*Figure 7.* Time evolution of the x-momentum field $m_x(x, y, t)$ comparing ground truth, FluxNet-LAP, FNO, and ResNet at six time steps. FluxNet-LAP maintains accurate predictions throughout, while ResNet develops high-frequency artifacts.

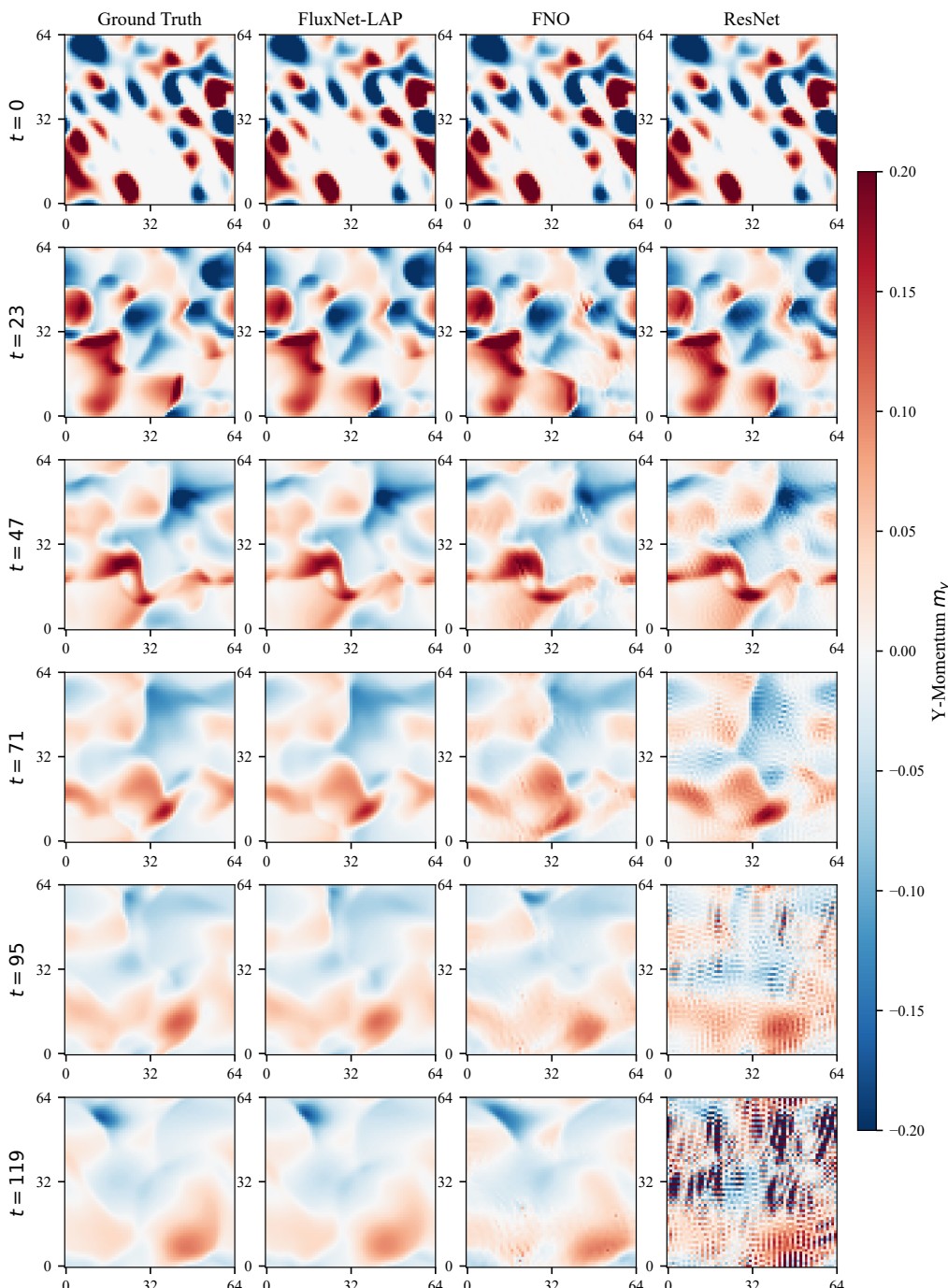

*Figure 8.* Time evolution of the y-momentum field $m_y(x, y, t)$ comparing ground truth, FluxNet-LAP, FNO, and ResNet at six time steps.

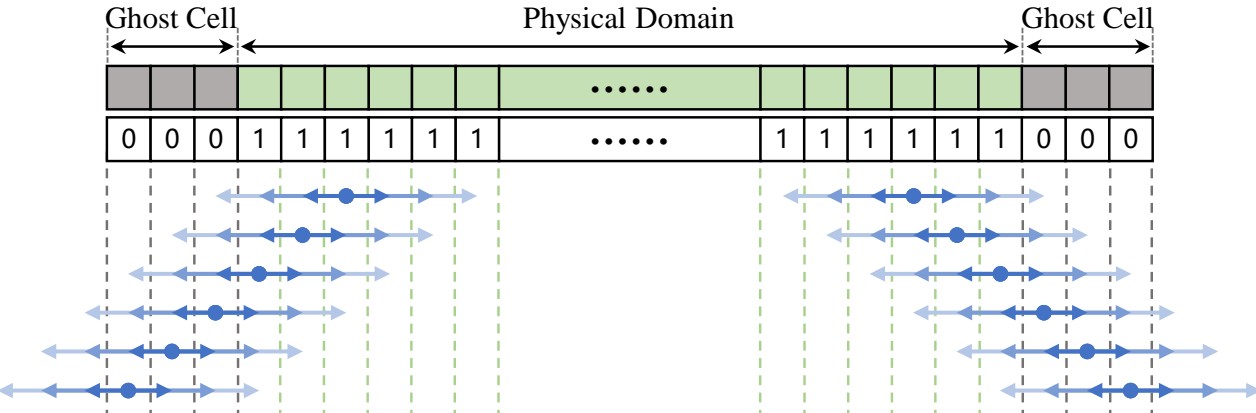

*Figure 9.* Schematic of the ghost cell boundary treatment for FluxNet-D under Dirichlet boundary conditions (1D, $R = 3$, outflow branch example). Ghost cells are padded on each side and filled with prescribed Dirichlet values. A binary identity channel (1 for interior, 0 for ghost) is appended to the input.

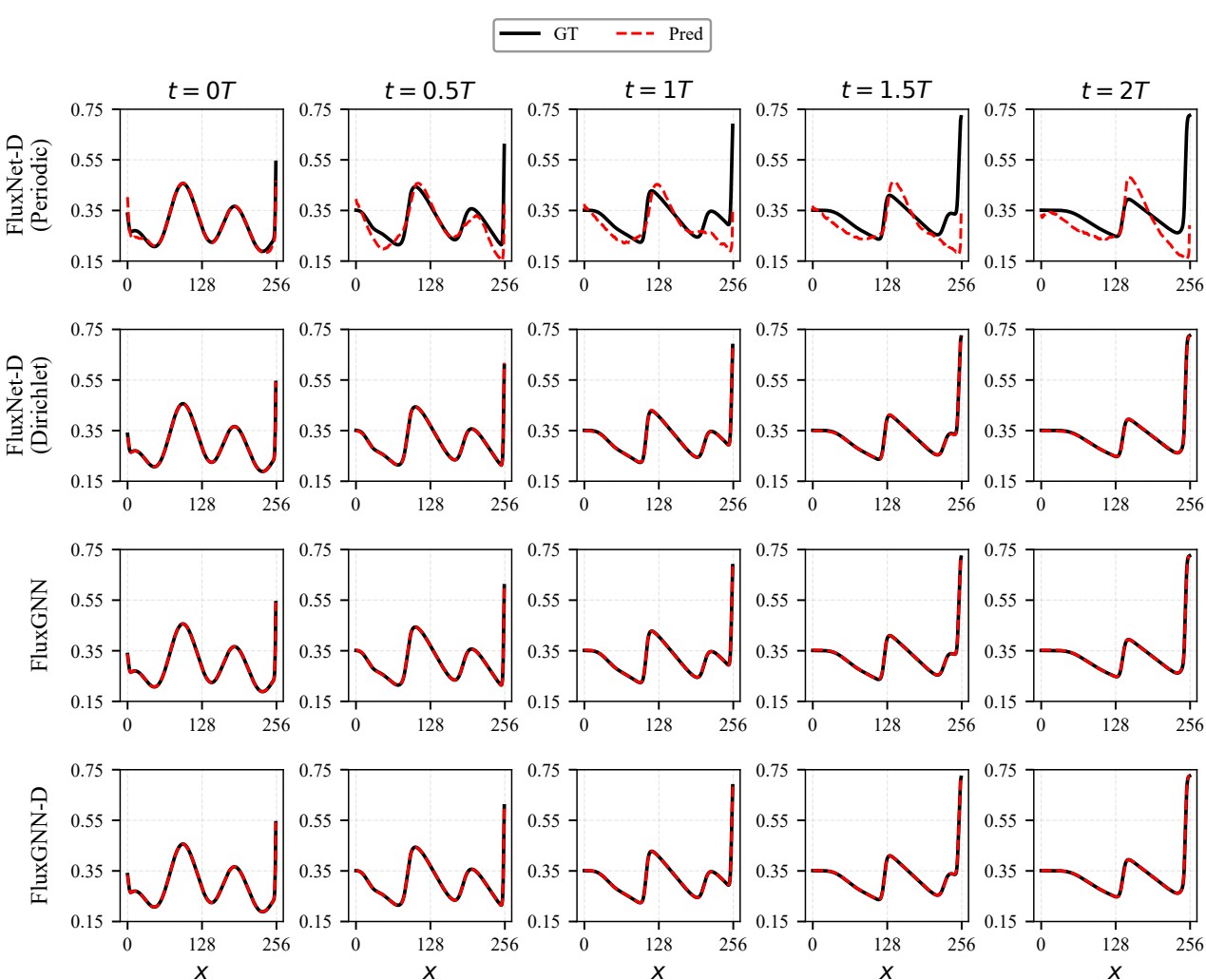

*Figure 10.* Rollout density profiles $\rho(x, t)$ at five time points ($t = 0T$ to $t = 2T$) for four methods on the Dirichlet traffic flow dataset ($\Delta t_{\text{model}} = 10\Delta t$). The periodic-only FluxNet-D fails catastrophically, while all three Dirichlet-aware methods produce accurate rollouts.

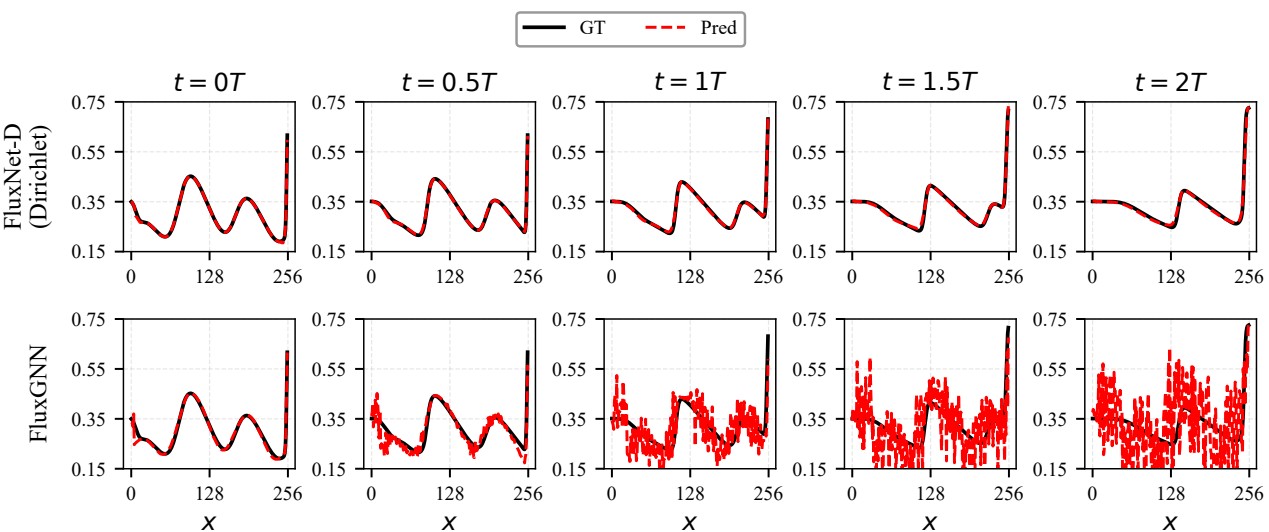

*Figure 11.* Rollout density profiles $\rho(x, t)$ at five time points comparing FluxNet-D (Dirichlet) and FluxGNN at the large temporal stride $\Delta t_{\text{model}} = 50\Delta t$. FluxNet-D maintains accurate tracking of ground truth, while FluxGNN diverges due to CFL limitations inherent in nearest-neighbor flux-rate prediction at large time steps.

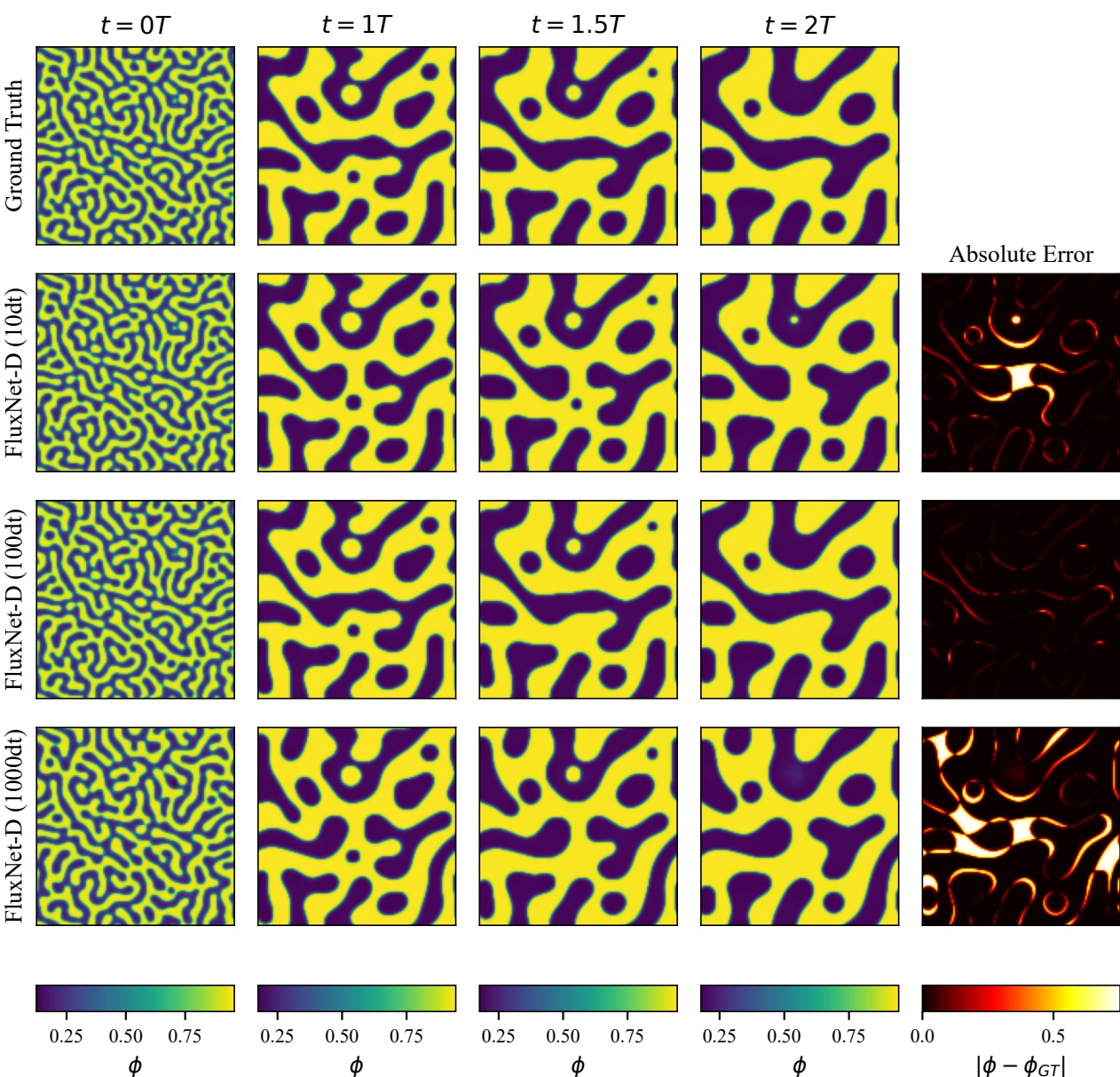

*Figure 12.* Concentration field $\phi(x, y, t)$ evolution comparing ground truth and FluxNet-D models trained with different time step sizes ($10\Delta t$, $100\Delta t$, $1000\Delta t$) at four time points ($0T$, $1T$, $1.5T$, $2T$). The rightmost column shows absolute prediction error at the final time $2T$.

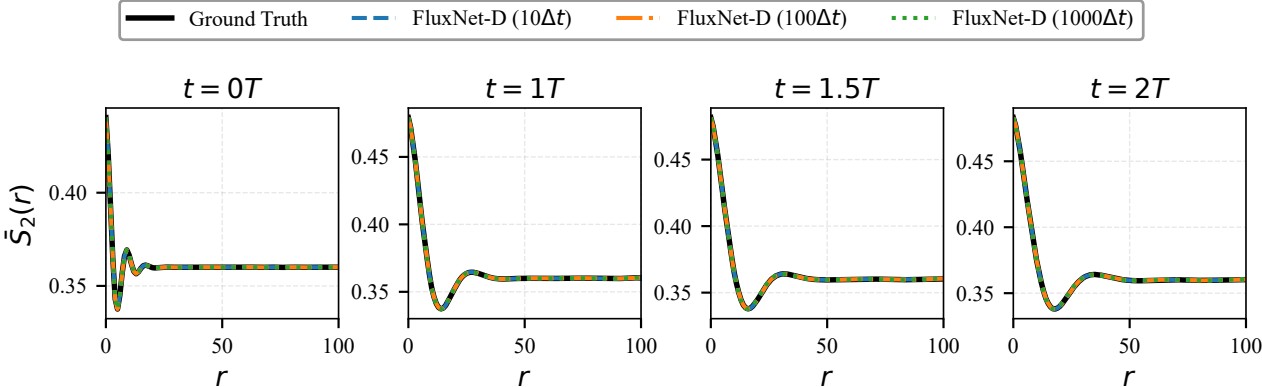

*Figure 13.* Radial two-point correlation function $\bar{S}_2(r)$ comparison between ground truth and FluxNet-D models at four time points: $0T$, $1T$, $1.5T$, and $2T$ ($2\times$ extrapolation). All models closely match the reference correlation functions.

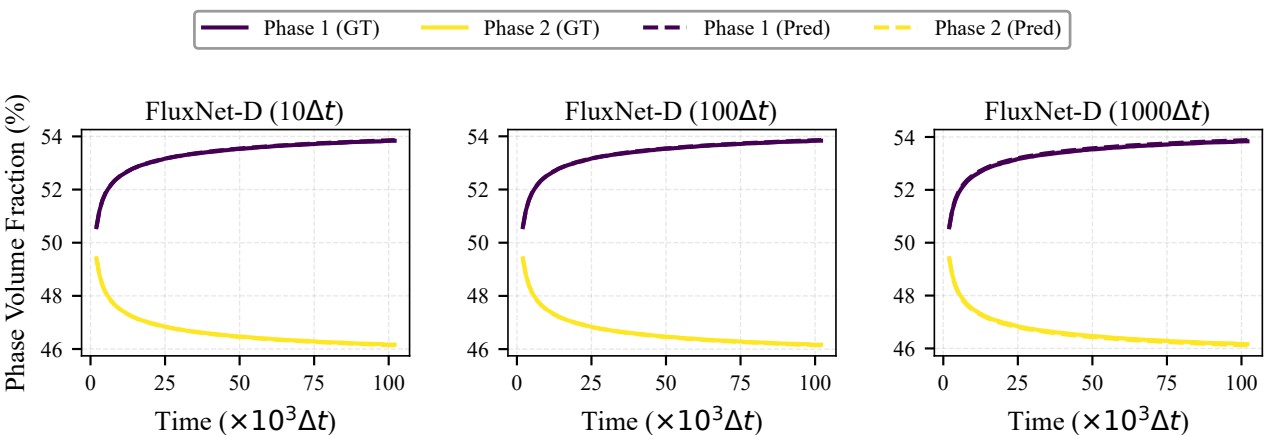

*Figure 14.* Phase volume fraction evolution during spinodal decomposition. Solid lines: ground truth; dashed lines: FluxNet-D predictions. All models accurately track the evolution of both high-concentration ($\phi \geq 0.6$) and low-concentration ($\phi < 0.6$) phase fractions.

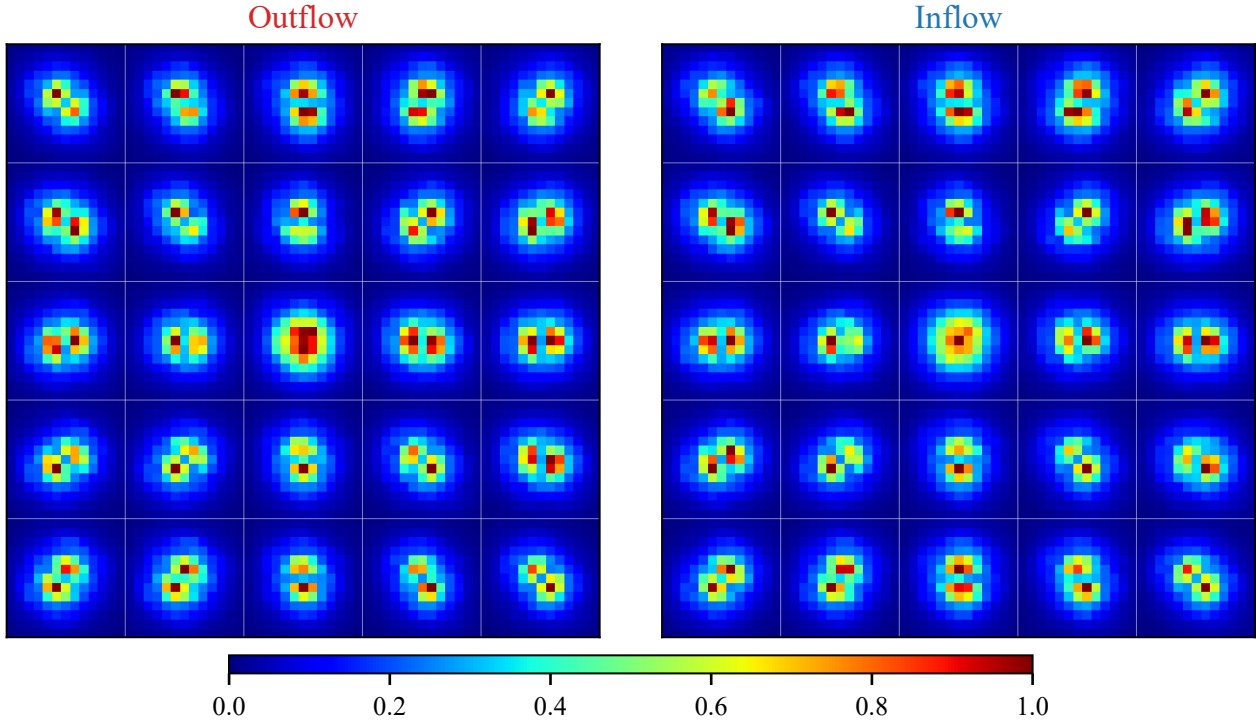

*Figure 15.* Effective receptive field analysis for FluxNet-D ($100\Delta t$). Left: outflow branch channels within the $5 \times 5$ neighborhood. Right: inflow branch channels. The larger ERF compared to the $10\Delta t$ model (Figure 4 in the main text) reflects extended transport range.

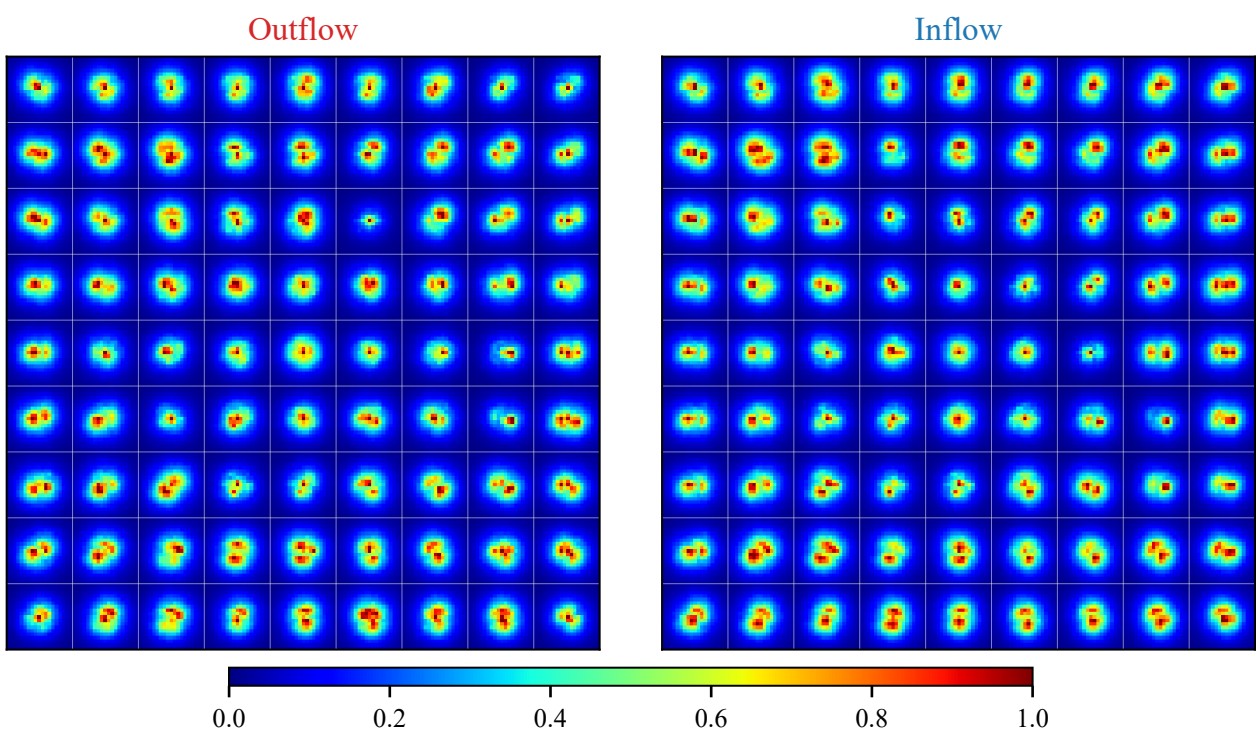

*Figure 16.* Effective receptive field analysis for FluxNet-D ($1000\Delta t$). Left: outflow branch channels within the $9 \times 9$ neighborhood. Right: inflow branch channels. This model exhibits the largest ERF among the three variants, consistent with extended spatial dependencies at the coarsest temporal resolution.

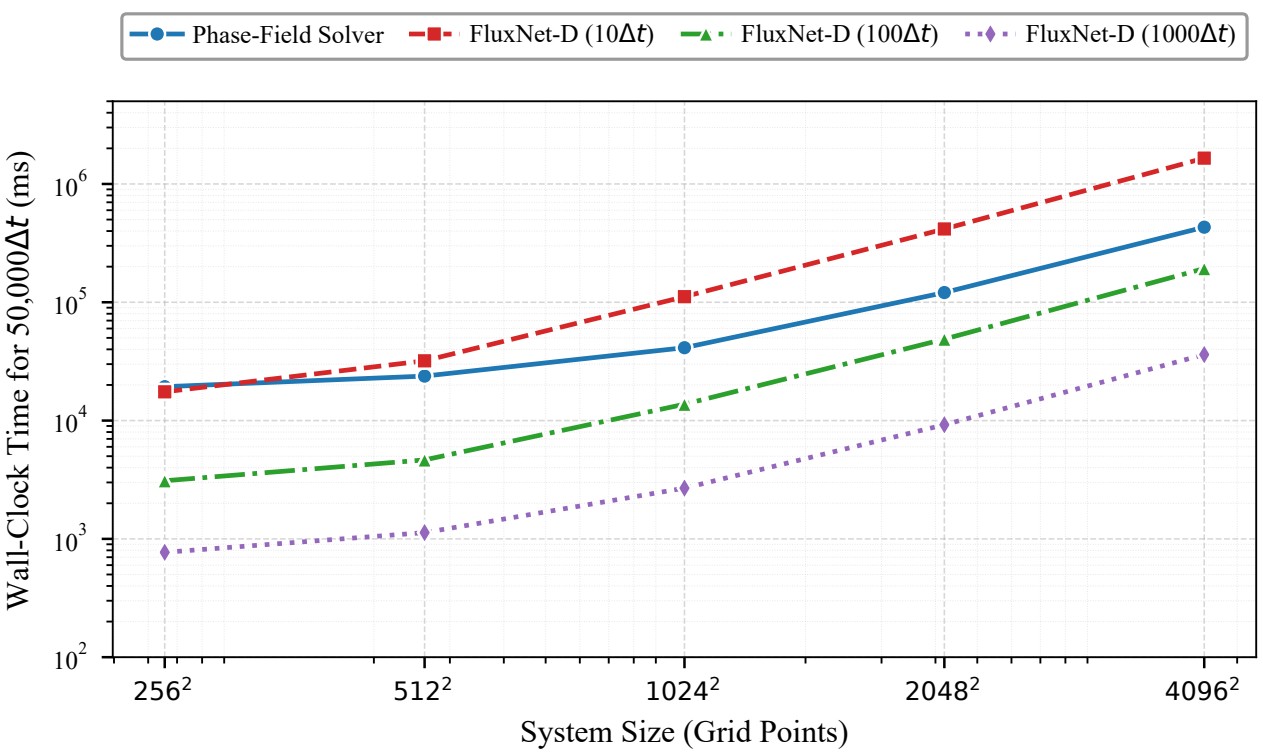

*Figure 17.* Wall-clock time for completing $50,000\Delta t$ of spinodal decomposition simulation. Comparison between the GPU-accelerated phase-field solver and FluxNet-D models at three temporal strides across system sizes from $256^2$ to $4096^2$ grid points. Both axes use logarithmic scales. The $1000\Delta t$ model achieves $17.3\times$ speedup at $1024^2$ resolution.

