# OpenReview forum: "FluxNet: Learning Capacity-Constrained Local Transport Operators for Conservative and Bounded PDE Surrogates"
_ICML.cc/2026/Conference — ICML 2026 regular_

### Official Review · Reviewer_oVF9 · 2026-02-16

**Soundness:** 3
**Presentation:** 4
**Significance:** 3
**Originality:** 3
**Overall Recommendation:** 4
**Confidence:** 3

**Summary:**

This paper introduces FluxNet, a family of neural PDE time-stepping architectures that achieve structural conservation and bound preservation by shifting the prediction target from next-state values to feasible local transport plans. The network outputs directional fluxes between neighboring grid cells, parameterized via a hierarchy of "transport heads" that, by construction, enforce increasingly strict physical constraints, and a subsequent conservative update tallies inflows/outflows to produce the next state. The approach is validated on four conservative PDE benchmarks (1D convection-diffusion, 2D shallow water, 1D traffic flow, 2D spinodal decomposition), demonstrating machine-precision conservation, reduced bound violations, and improved long-rollout stability over direct regression and soft-constraint baselines.

**Compliance With Llm Reviewing Policy:**

Affirmed.

**Final Justification:**

Rebuttal responses address all my concerns.

**Key Questions For Authors:**

1. Can you provide a direct comparison against FINN on at least one shared benchmark? FINN also learns flux functions for conservation laws and is the most natural competitor. A favorable comparison would significantly strengthen the contribution.

2. How do D-head violations behave during long rollout, do they accumulate or self-correct? The violation rates in Table 4 are per-snapshot. A time-series of cumulative violation magnitude would clarify whether the method is stable in the bound-violation sense over very long horizons. Evidence of self-correction would increase confidence in practical applicability.

3. What happens with non-periodic boundaries? Even a small experiment with Dirichlet conditions on one benchmark would demonstrate feasibility. Does the conservation proof break, and if so, how would boundary fluxes be handled Demonstrating non-periodic boundaries would substantially broaden the impact of this work.

4. What is the sensitivity to the $\varepsilon$ parameter in FluxNet-LAP?  How does performance change as $\varepsilon$ varies, especially in near-dry regions? This is a minor point but would improve reproducibility.

5. For the spinodal decomposition one-shot training: how sensitive is the model to the training trajectory's initial composition? Would a model trained at $c_0 = 0.60$ generalize to $c_0 = 0.50$ or $c_0 = 0.70$? Demonstrating composition generalization would strengthen the practical utility claim.

**Limitations:**

The authors have adequately discussed the limitations in Section 5 (Conclusion). They explicitly acknowledge: periodic boundary conditions, source-free conservation laws only, D-head's empirical rather than strict dual-bound guarantees, a fixed neighborhood size as a hyperparameter, and the absence of unstructured mesh support. This transparency is commendable and better than most submissions. However, the discussion would benefit from concrete suggestions for how to extend to non-periodic boundaries (the most important limitation for practical adoption).

**Strengths And Weaknesses:**

### Strengths:

- **S1**: Conservation is guaranteed by construction (Proposition 1), not by soft penalty or post-hoc correction. This is the right approach and the paper clearly demonstrates machine-precision conservation across all experiments.
- **S2**: Thorough and well-designed ablation studies (Tables 6, 7) systematically isolate the contribution of each component (head type, pressure gating, pushforward training, DCL), providing clear evidence for each design choice.
- **S3**: The spinodal decomposition experiment (Section 4.4) is exemplary, evaluating two-point correlation functions ($\bar{S}_2(r)$), phase volume fractions, and effective receptive fields goes well beyond standard pointwise error metrics and demonstrates genuine care for physical fidelity.
- **S4**: Transparent reporting of D-head limitations. Unlike many papers that would suppress violation statistics, FluxNet reports violation rates and conditional magnitudes as first-class metrics, building trust.
- **S5**: The backbone-head factorization enables fair comparison, all baselines share the same backbone, isolating the effect of the transport head parameterization.

### Weaknesses:

- **W1**: Missing direct comparison with FINN (Praditia et al., 2021; Karlbauer et al., 2022), the most natural competing flux-based approach. This is discussed in related work but never benchmarked, leaving a critical gap in the experimental evaluation.
- **W2**: Restriction to regular grids with periodic boundaries limits practical applicability. Many real scientific simulations involve complex geometries, unstructured meshes, and non-periodic boundary conditions. The paper does not discuss how the conservation proof or head designs would extend.
- **W3**: The D-head does not provide strict dual-bound guarantees, the averaged update from two branches can violate either bound. While the paper is honest about this, the violation rates of up to 7.09% (FluxNet-D FNO backbone, Table 4) are concerning for applications requiring strict physical bounds.
- **W4**: Computational cost is only reported for spinodal decomposition (Table 5). For the other three benchmarks, there is no wall-clock comparison against baselines, making it impossible to assess whether the structural constraints incur meaningful overhead.
- **W5**: The $\varepsilon$ regularization in the FluxNet-LAP momentum advection term (Section 3.5) is unspecified. Near dry regions ($h \approx 0$), the behavior of $m_i / (h_i + \varepsilon)$ is highly sensitive to this parameter, and no sensitivity analysis is provided.

---

> ### Author Rebuttal · Authors · 2026-03-31
>
> We sincerely thank Reviewer oVF9 for the thorough and constructive review, and for recognizing the strengths of our approach (S1–S5). We address each question below.
>
> **Q1&W1: Comparison with flux-based methods**
>
> We have conducted a direct comparison with FluxGNN [R1], the most directly comparable flux-based conservative surrogate. We chose FluxGNN over FINN because FINN is primarily a composable PDE learning framework rather than a dedicated flux prediction method—FluxGNN, grounded in finite volume principles, shares a more similar problem formulation with FluxNet. Full experimental details are in our response to Reviewer 5VJx (W1); our distinct contributions (bound preservation + CFL-free large-timestep prediction via integrated transport operators) are analyzed in our response to Reviewer Q9f2 (Q1&W1).
>
> > **Key results** (Table R1 in the supplement): At $\Delta t_{\text{model}} = 10\Delta t$, FluxGNN slightly outperforms FluxNet-D ($9.56$ vs. $14.1 \times 10^{-4}$) due to its more advanced backbone, but **FluxGNN-D** (FluxGNN backbone + our D-head) achieves the best MAE ($6.84 \times 10^{-4}$). At $\Delta t_{\text{model}} = 50\Delta t$, FluxGNN diverges (MAE $1540 \times 10^{-4}$) while FluxNet-D remains stable ($68.6 \times 10^{-4}$), confirming our large-timestep advantage.
>
> **Q2&W3: D-head violation behavior during long rollout**
>
> We acknowledge that violations from the averaged D-head update are generally **cumulative** rather than self-correcting—strict dual-bound satisfaction requires the two branches to agree exactly. However, two important observations support practical applicability:
>
> - **Violation magnitudes remain small**: conditional magnitudes are $O(10^{-3})$ (Table 4), meaning violations are low-amplitude excursions that do not trigger cascading instabilities (e.g., Gibbs-type oscillations near shocks) seen in unconstrained baselines.
> - **Practical rollout stability**: despite non-zero violation rates, FluxNet-D rollouts remain stable over 2× temporal extrapolation across all benchmarks without divergence.
>
> Achieving strict dual-bound guarantees requires the two branches to produce identical transport plans. This may involve novel training algorithms enforcing branch symmetry or lightweight iterative correction, which we identify as important future work.
>
> **Q3&W2: Non-periodic boundary conditions**
>
> Following the reviewer's suggestion, we implemented Dirichlet BC support via the ghost cell method—a standard computational physics boundary treatment where different BC types only change ghost cell assignment rules, leaving the transport mechanism unchanged. Full details are in our response to Reviewer 5VJx (Q1). Key result: FluxNet-D (Dirichlet) achieves **50× lower MAE** than the periodic version on a Dirichlet traffic flow benchmark (Table R1, Figure R2 in the supplement). Neumann/Robin conditions require only changing the ghost cell fill rule. Extension to unstructured meshes is discussed in our response to Reviewer Tjv9 (Q1).
>
> **Q4&W5: ε sensitivity in FluxNet-LAP**
>
> We tested $\epsilon$ from $10^{-2}$ to $10^{-8}$ using the trained model (Table R2 in the supplement). MAE decreases from $5.53 \times 10^{-3}$ at $\epsilon = 10^{-2}$ and **converges to $4.76 \times 10^{-3}$ for $\epsilon \leq 10^{-4}$**, remaining stable through $10^{-8}$. We use $\epsilon = 10^{-6}$. The insensitivity for $\epsilon \leq 10^{-4}$ indicates robustness—$\epsilon$ only matters in rare near-dry cells ($h \approx 0$), and any sufficiently small value yields equivalent performance.
>
> **Q5: Composition generalization**
>
> We evaluated the model trained at $c_0 = 0.60$ on unseen compositions $c_0 \in \{0.40, 0.50, 0.70, 0.80\}$ (Figure R4 in the supplement). All three FluxNet-D models produce visually realistic microstructures with correct morphology across all compositions. Two-point statistics errors for unseen compositions remain on the same order as for the training composition ($\sim 10^{-3}$), demonstrating robust composition generalization from single-trajectory training.
>
> **W4: Computational cost for other benchmarks**
>
> Wall-clock timing was reported only for spinodal decomposition because it uses a GPU-accelerated numerical solver, enabling fair GPU-vs-GPU comparison. The other three benchmarks use simple Python/CPU solvers where GPU neural network vs. CPU solver timing would be misleading. The spinodal experiment is the most informative for speedup analysis as it demonstrates the timestep-dependent efficiency tradeoff (Table 5).
>
> **[R1]** Graph Neural PDE Solvers with Conservation and Similarity-Equivariance. arXiv:2405.16183.
>
> **Supplement:** https://anonymous.4open.science/api/repo/04F4/file/m.pdf?v=79f822ce

---

> > ### Author Rebuttal · Reviewer_oVF9 · 2026-04-03
> >
> > Thanks for your rebuttal. This addresses all my concerns. I am keeping my score as-is.

---

> > > ### Author Response · Authors · 2026-04-03
> > >
> > > Thank you for such a careful and detailed review! Your questions helped us fill in several gaps and make the paper more complete. We appreciate your recognition of our responses and will incorporate everything into the final version.

---

### Official Review · Reviewer_Q9f2 · 2026-03-07

**Soundness:** 2
**Presentation:** 3
**Significance:** 2
**Originality:** 2
**Overall Recommendation:** 3
**Confidence:** 3

**Summary:**

FluxNet learns local transport between grid cells instead of directly predicting the next state, which guarantees exact conservation of mass through the update structure itself. Specialized transport heads enforce physical bounds: the L-head ensures lower bounds by limiting outflow, the U-head enforces upper bounds by constraining inflow, and the D-head handles doubly-bounded fields using two parallel branches with a dual consistency loss. Experiments on shallow water equations, traffic flow, and phase-field decomposition show that FluxNet achieves better long-term accuracy, maintains conservation at machine precision, respects physical bounds, and preserves statistical properties while enabling much larger time steps for substantial computational speedups.

**Compliance With Llm Reviewing Policy:**

Affirmed.

**Final Justification:**

FluxNet operates only on conserved variables $u$. However, for more complex or coupled systems, recovering primitive variables $\omega$ (e.g., velocity, pressure, or phase) from $u$ (i.e., PVR) is often ill-conditioned, highly nonlinear, or even singular.

In addition, the current experiments are limited to 1D/2D settings, where the mapping between conserved and primitive variables is relatively simple or explicitly defined. As a result, it remains unclear whether FluxNet can handle more challenging scenarios, such as 3D dynamics, complex-valued wavefunctions (where phase information may be lost), or systems with strong nonlinearities, where PVR is known to be a bottleneck.

Given these limitations, I am not yet convinced of the method’s general applicability, and therefore I maintain my overall recommendation of score 3.

**Key Questions For Authors:**

1. How does FluxNet differ fundamentally from existing neural flux approximation approaches, and what specific novelty does it introduce beyond the architecture design?
2. The flux-based transport formulation is physically appropriate for hyperbolic PDEs where information propagates at finite speed and solutions depend only on local neighborhoods, but for non-hyperbolic equations such as the heat equation where disturbances instantly affect the entire domain, does modeling transport only between adjacent cells align with the underlying physics?

**Limitations:**

yes

**Strengths And Weaknesses:**

Strengths:
1. Provides exact conservation guarantees at machine precision through the transport update structure, independent of network predictions.
2. Achieves long-term stability and accuracy compared to standard CNNs and FNOs across multiple benchmarks.

Weaknesses:
1. The core idea bears strong similarity to prior work [1] on approximating numerical fluxes using neural operators for hyperbolic conservation laws, yet that work is not cited or discussed, which raises concerns about novelty and situating the contribution within existing literature.
2. Currently limited to periodic boundary conditions and source-free conservation laws, restricting applicability to more general problems. The method also requires manual tuning of the fixed neighborhood size as a hyperparameter for each time step.
3. The framework is built around flux or transport formulations, which may not extend naturally to conservation laws involving higher-order derivatives or nonlinear combinations that cannot be readily expressed as local transport between grid cells.
4. The baseline comparisons are restricted to only ResNet and FNO architectures, which is a narrow set given the variety of existing neural PDE solvers and operator learning methods available for comparison.
5. The experimental comparison is limited to loss-based methods and does not include other structure-preserving or conservation-enforcing approaches [2, 3], making it difficult to assess how FluxNet performs relative to the broader landscape of methods designed to maintain physical constraints.

[1] Approximating Numerical Fluxes Using Fourier Neural Operators for Hyperbolic Conservation Laws. https://arxiv.org/abs/2401.01783

[2] Exactly conservative physics-informed neural networks and deep operator networks for dynamical systems. https://arxiv.org/abs/2311.14131

[3] Conservation-preserved Fourier Neural Operator through Adaptive Correction. https://arxiv.org/abs/2505.24579v1

---

> ### Author Rebuttal · Authors · 2026-03-31
>
> We thank the reviewer for the detailed and constructive feedback.
>
> **Q1&W1: Fundamental novelty beyond architecture design**
>
> We will cite [1] and FluxGNN [R1] in revision. Our contributions beyond *all* prior flux-based methods ([1], FINN, FluxGNN):
>
> **(a) Bound preservation.** Capacity-constrained heads (L/U/D) structurally enforce physical bounds (e.g., nonnegative density, concentrations in $[0,1]$)—absent from all existing flux-based surrogates.
>
> **(b) From flux rates to transport plans: CFL-free large-timestep modeling (key novelty).** The novelty lies in **what the network learns**, not merely the neighborhood size. FVM-inspired surrogates (FluxGNN, FINN, [1]) learn **instantaneous flux rates** at cell faces and compose them via explicit time integration, where the timestep explicitly multiplies the learned rate. The CFL condition arises as a stability requirement of this integration—not an inherent PDE property.
>
> FluxNet changes the semantics: instead of an instantaneous rate, the network outputs **cumulative transport amounts**—total conserved quantity redistributed from each cell to its neighbors over the *entire* interval. The state update is simply the current value minus total outflow plus total inflow. No temporal integration is performed; the timestep does not appear as a multiplier but is absorbed into the learned transport. The CFL condition is structurally eliminated. The extended neighborhood is a natural consequence: if material traverses $r$ cells during $\Delta t$, the transport plan needs neighbors at distance $r$, **decoupling temporal stride from spatial resolution**. This resolves the dilemma where tiny timesteps lose the surrogate's speed advantage (Table 5: the $10\Delta t$ model is *slower* than the GPU solver at $0.55\times$) while grid coarsening degrades accuracy. All structural guarantees (Props. 1–3) hold for any neighborhood size and any $\Delta t$, as properties of the transport-plan structure, not any integration scheme.
>
> > **Validation:** At $\Delta t_{\text{model}} = 50\Delta t$, FluxGNN (learning flux rates) **diverges** (MAE $1540 \times 10^{-4}$) while FluxNet-D stays stable ($68.6 \times 10^{-4}$).(Table R1, Figure R3.) The $1000\Delta t$ spinodal model achieves **17.3× speedup** preserving microstructural statistics. (Figure 3.)
>
> **Q2&W3: Non-hyperbolic PDEs, higher-order derivatives, nonlinear combinations**
>
> Two clarifications: **(1)** As in (b), FluxNet learns cumulative transport with configurable extended neighborhoods, not only adjacent-cell instantaneous fluxes. **(2)** Our spinodal decomposition (Section 4.4) **already validates FluxNet on a non-hyperbolic, higher-order, nonlinear PDE.**
>
> **General principle.** Any source-free conservative system can be written as $\partial u / \partial t = -\nabla \cdot \mathbf{J}$, regardless of $\mathbf{J}$'s complexity. FluxNet learns the cumulative effect of $\mathbf{J}$ through local grid exchanges—complexity resides in $\mathbf{J}$, not the transport update structure.
>
> **Cahn-Hilliard equation.** Eq. 18 is a **fourth-order parabolic PDE** whose flux contains a nonlinear term (via $f''(c)$ containing $\ln[c/(1-c)]$) and third-order spatial derivatives. Together they yield a fourth-order PDE. Our results (Table 5, Figures 3–4) directly confirm FluxNet handles such physics.
>
> **Heat equation.** It fits naturally as $\partial T / \partial t = \nabla \cdot (\alpha \nabla T)$. Although mathematically it has infinite propagation speed, for finite $\Delta t$ the Green's function decays exponentially, making transport localized—precisely why traditional solvers also use local stencils.
>
> The reviewer may have in mind **elliptic** PDEs (Laplace/Poisson), where disturbances genuinely affect the entire domain instantaneously. These contain **no** $\partial u / \partial t$—they describe steady-state equilibria, not time evolution. As a time-stepping operator (Eq. 1), FluxNet targets evolutionary problems; steady-state equations are outside our scope. We will clarify this in revision.
>
> **W2: Periodic BC limitation**
>
> Extended to Dirichlet BCs via ghost cells—see Reviewer 5VJx (Q1). Key result: **50× MAE improvement** (Table R1).
>
> **W4&W5: Baseline breadth**
>
> Unlike post-hoc methods ([2],[3]), FluxNet guarantees conservation by construction. Added FluxGNN [R1] as flux-based baseline (Reviewer 5VJx, W1). FluxGNN-D (our D-head on FluxGNN backbone) achieves best MAE in Table R1, confirming modularity.
>
> **[R1]** Graph Neural PDE Solvers with Conservation and Similarity-Equivariance. arXiv:2405.16183.
>
> **Supplement:** https://anonymous.4open.science/api/repo/04F4/file/m.pdf?v=79f822ce

---

> > ### Author Rebuttal · Reviewer_Q9f2 · 2026-04-03
> >
> > I thank the authors for the detailed rebuttal and the additional experiments. While the transition from "flux rates" to "transport plans" offers some novelty, several fundamental concerns regarding the structural limitations and the physical consistency of the framework remain.
> >
> > 1. Pathological Bottleneck of Primitive Variable Recovery (PVR)
> >
> > FluxNet primarily updates conserved quantities. However, in most physical systems, the transport $J$ depends on primitive variables (e.g., velocity, pressure, or phase), which must be recovered from these conserved quantities. This recovery process involves solving an inverse functional system $w = f^{-1}(u)$ that is often highly non-linear, numerically ill-conditioned, or even non-uniquely defined (e.g., at phase transition boundaries or wetting-drying fronts).This issue is even more pronounced in complex-valued domains (e.g., the Schrödinger equation), where the conserved quantity is a scalar (density), but the underlying primitive variables are complex numbers. Since the "transport plan" cannot inherently capture the phase information required to calculate future fluxes, the framework's applicability to such higher-order or wave-like systems is theoretically questionable.
> >
> > The necessity of using a manually engineered "LAP-head" for SWE further suggests that FluxNet is not an "out-of-the-box" general operator, but rather a specialized tool that requires significant expert-designed architectural priors to handle numerical singularities in the primitive recovery process.
> >
> > 2. Over-Simplified Experimental Validation
> >
> > The current benchmarks are restricted to 1D and 2D systems with relatively simple, explicit relationships between fluxes and conserved variables. The experiments lack validation on systems with complex constitutive laws or high-degree non-linearity, such as:
> >
> >   Relativistic Fluids: Where PVR is a classic numerical bottleneck requiring iterative solvers.
> >
> >   Reactive Flows: Where fluxes are exponentially sensitive to temperatures recovered from energy densities.
> >
> >   Non-linear Elasticity: Where the stress-strain relationship may involve hysteresis or multi-valued mappings.
> >
> > Overall, I maintain my original score.

---

> > > ### Author Response · Authors · 2026-04-04
> > >
> > > We sincerely thank the reviewer for the continued and thought-provoking exchange. The back-and-forth—especially the push to articulate what sets transport plans apart from flux rates—has helped us sharpen the core contribution, and we are carrying this improved framing into the revision. We hope the following discussions can address the new concerns raised by the reviewers:
> > >
> > > **On Primitive Variable Recovery (PVR).** PVR is fundamentally an algorithmic bottleneck of classical CFD: traditional solvers must execute $U \to W$ as a standalone algebraic inversion (typically via Newton–Raphson iterations), which crashes—diverges or yields NaNs—at phase boundaries or dry fronts due to mathematical singularities. In contrast, neural surrogates (including FluxNet, FNO, and FluxGNN) map conserved variables to future states or fluxes via high-dimensional, regularized latent representations. Because the explicit, step-by-step algebraic inversion is structurally bypassed, the classical runtime crashes associated with PVR simply do not arise. The underlying physical complexity (e.g., extreme nonlinearity) manifests instead as a learning difficulty in extreme regimes. Overcoming this requires richer data or model capacity—a universal challenge across scientific ML rather than a structural impossibility of transport plans.
> > >
> > > **On the Schrödinger equation.** The Schrödinger equation $i\hbar\,\partial\psi/\partial t = H\psi$ evolves a complex wavefunction. While probability conservation $\partial|\psi|^2/\partial t + \nabla\cdot\mathbf{J}=0$ is a derived identity, it is not a closed conservation law: the probability current $\mathbf{J}$ depends on both amplitude and phase. It therefore falls outside our targeted scope of closed, source-free conservation laws. However, under the Madelung transformation (quantum hydrodynamics), the system rigorously decomposes into a continuity equation for $\rho=|\psi|^2$ coupled with a quantum Hamilton–Jacobi equation for the phase—structurally analogous to our SWE benchmark where depth couples with momentum. FluxNet could govern the density conservation step while a companion module evolves the phase. We view this as a concrete and promising future extension rather than a theoretical barrier.
> > >
> > > **On the three proposed benchmark systems.**
> > > - **(a) Relativistic hydrodynamics** is governed by $\partial_\mu T^{\mu\nu}=0$ and $\partial_\mu(nu^\mu)=0$—source-free conservation laws that fall within FluxNet's problem class in principle. The main challenge lies in the learning difficulty posed by the extreme nonlinearity of the equation of state and Lorentz-factor coupling, rather than any structural incompatibility with transport plans. We regard this as a concrete and worthwhile future direction.
> > > - **(b) Reactive flows** involve species equations $\partial(\rho Y_k)/\partial t + \nabla\cdot(\rho Y_k \mathbf{v}) = \omega_k$ with chemical source terms $\omega_k \neq 0$, which fall outside our source-free scope as acknowledged in Section 5. We note that the "exponential sensitivity to temperature" the reviewer highlights refers to Arrhenius reaction rates $\omega_k \sim \exp(-E_a/RT)$—these are source terms rather than transport fluxes. Extending FluxNet to handle source terms via operator splitting (FluxNet handles transport; a separate module handles reactions) is a natural next step that we will discuss in the revision.
> > > - **(c) Nonlinear elasticity with hysteresis** involves path-dependent constitutive laws where current stress depends on loading history. The reviewer rightly notes that single-frame autoregressive prediction cannot capture such dependence. However, conditioning on multiple past frames is a well-established technique in autoregressive modeling (used in video prediction, turbulence forecasting, etc.), and FluxNet's transport structure is fully compatible with multi-frame input. This represents a promising extension for autoregressive surrogates broadly, including FluxNet.
> > >
> > > Of the three, (a) is directly within our problem class and awaits future experimental validation. (b) and (c) require extensions to source terms and history-dependent constitutive laws respectively—both go beyond this paper's scope but represent promising directions where FluxNet's conservation structure could play a meaningful role.
> > >
> > > **On the LAP-head.** Incorporating domain-specific inductive biases is standard practice in scientific ML. The LAP-head decomposes SWE momentum transport into two physically distinct components: advective momentum carried by inflowing water mass, and pressure-gradient-driven momentum—matching the two mechanisms in the coupled $(h, hu, hv)$ system. This is a physics-informed design choice rather than a workaround for numerical singularity. The generic L/U/D heads require no expert design and succeed on three benchmarks spanning hyperbolic, parabolic, and fourth-order nonlinear PDEs. That the framework accommodates both generic and specialized heads demonstrates its extensibility.

---

### Official Review · Reviewer_5VJx · 2026-03-12

**Soundness:** 3
**Presentation:** 4
**Significance:** 4
**Originality:** 3
**Overall Recommendation:** 5
**Confidence:** 4

**Summary:**

The authors propose an intrinsically conservative autoregressive neural operator scheme: the state of a field variable at time t_1 is used to predict its state at time t_2 by predicting the fluxes between grid cells on a quadrilaterally-meshed domain with periodic boundary conditions. The authors also propose a clever use of “heads” to predict the flux in both directions and then apply a scheme that is reminiscent of flux-limiters from traditional CFD in order to preserve boundedness. They achieve particularly impressive results on a 1D traffic flow problem where the network accurately reproduces sharp shocks with no Gibbs phenomena at the boundary.

**Compliance With Llm Reviewing Policy:**

Affirmed.

**Final Justification:**

I have updated the score to an accept (5) based on the additional results presented in the rebuttal. The main weaknesses of the first iteration were the restriction to periodic BCs, lack of comparison to and discussion of comparable methods. Each of these points was addressed in the rebuttal and demonstrated comparable performance for short horizons and improved stability over longer horizons, which while not comprehensive is sufficient to support the novelty and significance of this method. The method was already well formulated and presented, the response clarified the novelty and improved the empirical support, I think this would be a useful and timely contribution.

**Key Questions For Authors:**

Questions:
Q1. What would be needed for extensions to non periodic boundary conditions and unstructured domains?

**Limitations:**

yes

**Strengths And Weaknesses:**

Strengths:

S1. The presentation is very good.

S2. The results are visually impressive in addition to being numerically good. It is not easy to suppress the ringing phenomena at shock boundaries in traditional numerical solvers without flux-limiting schemes. It’s impressive they were able to do so with a relatively simple approach while still maintaining conservation to machine precision.

Weaknesses:

W1. I would have liked to have seen a comparison against a finite-volume based approach like FINN (which they do cite, but don’t compare to). This work seems like an extension to finite volume methods, not a replacement for FNOs, and so the comparisons they do make (like post-hoc projection or output squashing) seem like comparisons to methods they *should* outperform, not to the relevant SOTA. While I do suspect they would beat FINN, it would be nice to see.

---

> ### Author Rebuttal · Authors · 2026-03-31
>
> We sincerely thank the reviewer for the positive evaluation and for recognizing the shock-capturing quality (S2) and overall presentation (S1). We address each point below.
>
> **Q1: Extension to non-periodic BCs and unstructured domains**
>
> ***(a) Non-periodic boundary conditions.*** We have implemented and validated Dirichlet BC support via the **ghost cell method**—a standard boundary treatment in computational physics where different BC types only change ghost cell value assignment rules, leaving the internal transport mechanism entirely unchanged.
>
> **Method.** Given a physical domain $\Omega = \{1, \ldots, N\}$ with prescribed boundary values $u_L, u_R$, we pad $R = (K-1)/2$ ghost cells on each side ($K$: stencil width), filled with the Dirichlet values. The backbone (with circular padding replaced by replicate padding) and flux head operate on the extended domain $\Omega_{\text{ext}}$. A binary **identity channel** (1 for interior, 0 for ghost) is appended to the input, enabling the network to learn boundary-aware flux behavior without modifying the transport mechanism. The transport update proceeds identically on $\Omega_{\text{ext}}$; only interior positions are extracted as the next state. (See Figure R1 in the supplement.)
>
> **Conservation becomes flux-balance.** On the extended domain, the transport update preserves total mass exactly (Proposition 1). Partitioning into interior and ghost contributions yields:
>
> $$\sum_{i \in \Omega} u_i^{t+1} = \sum_{i \in \Omega} u_i^{t} + \Phi_{\text{boundary}}^t$$
>
> where $\Phi_{\text{boundary}}^t = -\sum_{g \in G} \Delta u_g^t$ is the net boundary flux. Domain mass changes *only* through boundary fluxes—a physically correct flux-balance identity.
>
> **Results.** On 1D traffic flow with Dirichlet BCs (Table R1, Figure R2 in the supplement), FluxNet-D (Dirichlet) achieves MAE of $14.1 \times 10^{-4}$—a **50× improvement** over the periodic-only version ($709.0 \times 10^{-4}$), confirming successful Dirichlet modeling.
>
> **Generality.** Dirichlet is the most demanding non-periodic BC type—it directly introduces external values that challenge transport feasibility constraints. Neumann and Robin conditions follow naturally by changing *only* the ghost cell assignment rule (gradient-based extrapolation or linear combination), requiring **no modification** to the transport heads or conservation mechanism.
>
> ***(b) Unstructured domains.*** We discuss extension to irregular meshes via graph-based extended transport neighborhoods in our response to Reviewer Tjv9 (Q1).
>
> **W1: Comparison with flux-based conservative surrogates**
>
> All four reviewers rightly requested comparison with flux-based methods. We conducted a thorough re-examination of the landscape to identify the most appropriate baseline.
>
> **Why FluxGNN rather than FINN.** Our related work discussion of FINN was overly compressed, creating the impression that FINN and FluxNet solve the same problem. FINN is primarily a composable PDE learning framework that decomposes equations into constituent functions (diffusion, advection, reaction); flux prediction is one component, not the central contribution. FluxGNN [R1], by contrast, is also grounded in finite volume principles and directly predicts inter-cell fluxes for conservative state updates—sharing the most similar problem formulation with FluxNet. This makes FluxGNN the most natural competitive baseline for our work, and we therefore selected it as our primary point of comparison. We will clarify this distinction in the camera-ready version.
>
> **Results** (Table R1, Figures R2–R3 in the supplement):
> We implemented FluxGNN on regular grids for the Dirichlet traffic flow benchmark.
>
> - At $\Delta t_{\text{model}} = 10\Delta t$: FluxGNN achieves slightly lower MAE ($9.56 \times 10^{-4}$) than FluxNet-D ($14.1 \times 10^{-4}$), attributable to FluxGNN's more advanced backbone; FluxNet uses a standard ResNet yet achieves comparable performance.
>
> > **FluxGNN-D** (our D-head on FluxGNN's backbone) achieves the **best MAE** ($6.84 \times 10^{-4}$), demonstrating that our transport heads are **modular and complementary**—they plug into existing flux architectures for immediate improvement.
>
> - At $\Delta t_{\text{model}} = 50\Delta t$: **FluxGNN diverges** (MAE $1540 \times 10^{-4}$) while FluxNet-D remains stable ($68.6 \times 10^{-4}$), demonstrating FluxNet's fundamental large-timestep capability.
>
> This work extends the finite volume paradigm from instantaneous flux rates to cumulative transport plans, and the resulting CFL-free large-timestep capability is the key advancement. Detailed analysis is provided in our response to Reviewer Q9f2 (Q1&W1).
>
> **[R1]** Graph Neural PDE Solvers with Conservation and Similarity-Equivariance. arXiv:2405.16183.
>
> **Supplement:** https://anonymous.4open.science/api/repo/04F4/file/m.pdf?v=79f822ce

---

> > ### Author Rebuttal · Reviewer_5VJx · 2026-04-01
> >
> > The author response thoroughly addresses the concerns raised above, particularly in the more general extension and additional comparisons, which substantially improve the paper.

---

> > > ### Author Response · Authors · 2026-04-03
> > >
> > > Thank you for the encouraging and insightful review! Your question about non-periodic boundaries pushed us to implement and validate Dirichlet BC support, and the suggestion to compare with FVM-based baselines led to the FluxGNN experiments. Both are solid additions to the paper — glad these came up during review.

---

### Official Review · Reviewer_Tjv9 · 2026-03-13

**Soundness:** 3
**Presentation:** 3
**Significance:** 3
**Originality:** 2
**Overall Recommendation:** 4
**Confidence:** 2

**Summary:**

The submission enforces conservation laws in PDE surrogate modeling, which improves rollout prediction stability. FluxNet predicts fluxes between neighboring grid cells to guarantee global conservation laws by construction, and has specifically designed heads to ensure boundedness of certain quantities. The model is validated across four different benchmarks to demonstrate its effectiveness.

**Compliance With Llm Reviewing Policy:**

Affirmed.

**Final Justification:**

The rebuttal addressed my concerns. The added experiments improved the soundness of the evaluation, and the authors claim that this work can be readily extended to irregular grids, which will strengthen the potential impact. Therefore, I updated my score to weal accept.

**Key Questions For Authors:**

1. Can the proposed method be applied to irregular grids represented by, e.g., a graph? This may increase the generality and impact of the proposed method.
2. See weakness 2. Other than preserving boundedness, what is the exact improvement of FluxNet, compared to the existing FINN and other methods mentioned in subsection 2.2?

**Limitations:**

Yes

**Strengths And Weaknesses:**

Strengths:
1. FluxNet proposes a new architecture to preserve the conserved quantities and boundedness. It has the potential to resolve the vital problem of long rollouts in neural PDE solvers.
2. The proposed method is well-motivated and enforces conservation laws successfully, as evaluated on several benchmarking systems.


Weaknesses:
1. The method is limited to conservative systems, regular grids, source-free assumption, and overall, the prior knowledge of the conserved properties in the system. This limits the generality of the method, but it is not necessarily a problem.
2. I am not familiar with the field of conservation law embedded neural PDE solvers. However, the submission claims its contribution lies in modeling the flux to preserve conservation and boundedness, while it mentions several related works that also embed the flux structure. The contribution would be more convincing and clear if the relation between FluxNet and other conservation-based neural surrogates (such as FINN mentioned in the related work section) were more clearly discussed.
3. Related to weakness 2, the empirical study does not compare FluxNet with flux-based conservative neural surrogates mentioned in section 2.2. I am not sure if those are applicable, but they seem to be stronger than soft constraints and post-hoc projection, which are the only baselines included in the experiments. Additionally, the experiments only included relatively simple 1D and 2D cases, leaving the scalability of FluxNet in more complex systems unclear.

---

> ### Author Rebuttal · Authors · 2026-03-31
>
> We thank the reviewer for recognizing FluxNet's potential to resolve the long-rollout problem and for the constructive feedback. We address each concern below.
>
> **Q1: Extension to irregular grids**
>
> This is an important direction we plan to pursue. FluxGNN [R1], a finite-volume-inspired GNN method, has demonstrated that predicting inter-cell fluxes on irregular meshes can preserve conservation. However, FluxGNN learns instantaneous face flux rates between immediate mesh neighbors, inheriting CFL stability constraints that restrict it to small time steps (detailed in our response to Reviewer Q9f2, Q1&W1).
>
> Our LBM-inspired transport neighborhood concept naturally extends to unstructured meshes: by connecting each node to all nodes within a physical radius (beyond immediate mesh neighbors), one defines extended exchange neighborhoods analogous to our regular-grid stencils. Our transport heads (L/U/D) operate on *any* neighborhood structure—they only require a set of neighbors with associated outflow/inflow parameterization. Specifically:
>
> - Shift-based flux accumulation is replaced by message-passing aggregation over extended neighborhoods
> - Edge features (distance, orientation) inform directional flux prediction
> - The conservation proof (Proposition 1) holds on any symmetric graph neighborhood
>
> This can be viewed as extending FluxGNN from nearest-neighbor instantaneous flux rates to multi-hop capacity-constrained transport plans, enabling large-timestep prediction on unstructured meshes. We will include this discussion in the camera-ready version.
>
> **Q2&W2: Exact improvement over FINN and flux-based methods**
>
> We acknowledge that our related work discussion of FINN was too compressed, which may have obscured the distinction. FINN's primary contribution is a composable PDE learning framework that decomposes equations into constituent functions (diffusion, advection, reaction); flux learning is one component, not the central focus. Beyond boundedness preservation, FluxNet's key advances over prior FVM-inspired flux-based methods (e.g., FluxGNN [R1]) are:
>
> - **CFL-free large-timestep capability**: FVM-inspired methods learn instantaneous face flux rates between nearest neighbors, inheriting CFL stability constraints from explicit time integration. FluxNet shifts the learned quantity from instantaneous rates to cumulative transport amounts, eliminating the CFL condition and decoupling temporal stride from spatial resolution. Detailed analysis is in our response to Reviewer Q9f2 (Q1&W1).
> - **Modularity of transport heads**: Our transport heads are transferable—grafting the D-head onto FluxGNN's backbone (FluxGNN-D) further improves performance (see response to Reviewer 5VJx, W1).
>
> **W1: Scope limitations**
>
> FluxNet intentionally targets conservation-law PDEs—a ubiquitous class in physics encompassing mass, momentum, and energy transport. The source-free assumption holds for many important systems (e.g., mass conservation in fluid dynamics, phase-field evolution).
>
> **W3: Comparison with flux-based baselines and experiment complexity**
>
> We conducted new experiments comparing with FluxGNN [R1] on a Dirichlet-boundary traffic flow dataset (see response to Reviewer 5VJx, W1). Key findings: (1) FluxNet with a simple ResNet backbone achieves accuracy comparable to FluxGNN's more advanced architecture; (2) our D-head is complementary to existing architectures (FluxGNN-D achieves the best accuracy); (3) FluxNet-D uniquely maintains stability at large time steps ($\Delta t_{\text{model}} = 50\Delta t$) where FluxGNN diverges.
>
> Regarding complexity: our benchmarks span traffic flow with shocks and rarefaction waves (as Reviewer 5VJx noted: "not easy to suppress ringing phenomena at shock boundaries"), the 4th-order parabolic Cahn-Hilliard equation, and coupled shallow water equations with dry regions—these are genuinely challenging problems. Success in both 1D and 2D with diverse physics suggests straightforward extension to 3D.
>
> **[R1]** Graph Neural PDE Solvers with Conservation and Similarity-Equivariance. arXiv:2405.16183.
>
> **Supplement:** https://anonymous.4open.science/api/repo/04F4/file/m.pdf?v=79f822ce

---

> > ### Author Rebuttal · Reviewer_Tjv9 · 2026-04-02
> >
> > I thank the authors for the detailed reply. It has addressed my concerns. I have updated my score to reflect this.

---

> > > ### Author Response · Authors · 2026-04-03
> > >
> > > Thanks for the helpful review and for confirming that our rebuttal addressed your concerns! Your questions inspired us to think more carefully about extension to irregular grids and to run the FluxGNN comparison — both made the paper stronger. We'll update the related work discussion in the camera-ready to make these distinctions clearer.

---

### Decision · Program_Chairs · 2026-04-30

**Decision:**

Accept (regular)

**Comment:**

This paper proposes FluxNet, a neural PDE time-stepping framework that predicts feasible local transport plans instead of directly regressing the next state. This design gives the method a clear and meaningful advantage: conservation is enforced by construction, and boundedness can also be incorporated structurally through the transport heads. The paper is well motivated, and the main idea is both technically interesting and practically relevant for long-horizon rollouts of conservative systems. As highlighted by several reviewers, the method achieves machine-precision conservation, shows strong long-term stability, and performs particularly well on challenging settings such as traffic flow with sharp shocks and spinodal decomposition, where physical consistency matters beyond pointwise error. The rebuttal also strengthened the paper substantially by adding comparisons to a more relevant flux-based baseline, clarifying the distinction from prior finite-volume-inspired neural surrogates, and demonstrating extensions to Dirichlet boundary conditions. Overall, the submission now presents a clearer and better-supported case for its contribution.

The paper does have limitations. In particular, as raised by Reviewer Q9f2, the experiments do not yet cover more complex 3D dynamics or settings in which recovering primitive variables becomes especially difficult. This is an important direction. However, in my judgment, this concern does not outweigh the strengths of the current paper. The method and experiments are self-contained, and the current empirical results are sufficient to demonstrate the effectiveness of the proposed transport-plan formulation within the intended problem class. Rather than a reason for rejection, the missing 3D validation is better viewed as an important future direction that should be discussed more explicitly in the final version. I therefore recommend acceptance. The paper makes a meaningful contribution to conservative neural PDE surrogates, and its structural approach to conservation and boundedness is likely to be useful to the community.